# Beyond wind speed: Integrating oceanic indices and time-lagged features for superior wind energy prediction

Namal Rathnayake[1,2]*, Mahesh Yadev[3], Jeevani Jayasinghe[4], Upaka Rathnayake[5], Masashi Minamide[1], Yukinobu Hoshino[6]

1 Graduate School of Engineering, Faculty of Engineering, University of Tokyo, Hongo, Tokyo, Japan, 2 Advanced Institute for Marine Ecosystem Change (WPI-AIMEC), Japan Agency for Marine-Earth Science and Technology (JAMSTEC), Yokohama, Japan, 3 Ministry of Water Supply, Irrigation and Energy, Koshi Province, Nepal, 4 Department of Electronics, Faculty of Engineering, Wayamba University, Kurunegala, Sri Lanka, 5 Department of Civil Engineering and Construction, Faculty of Engineering and Design, Atlantic Technological University, Sligo, Ireland, 6 School of Systems Engineering, Kochi University of Technology, Kami City, Kochi, Japan

* namal@hydra.t.u-tokyo.ac.jp, namalr@jamstec.go.jp

## Abstract

Accurate wind energy forecasting is critical for integrating wind power into electrical grids due to its inherent variability and uncertainty. This study introduces a systematic framework that integrates large-scale oceanic climate indices and time-lagged features with advanced machine-learning models to enhance short-term wind power prediction. We evaluate four experimental configurations: (A) a baseline using only wind speed; (B) wind plus contemporaneous indices; (C) the addition of 1–12 month lags for both wind and index variables; and (D) MRMR-based feature selection applied to the full lagged set. A comprehensive benchmark using 25 state-of-the-art models is conducted on monthly data from the Pawan Danavi wind farm in Sri Lanka (2015–2019). Results reveal that raw indices alone can degrade forecast accuracy, while incorporating lagged features significantly reduces RMSE and enhances $R^2$. MRMR pruning of the 156 lagged predictors distills the set to three key variables: current wind speed, a nine-month lag of the Atlantic Meridional Mode, and a six-month lagged wind speed. This yields a minimum RMSE of $\approx 50$ MWh and $R^2 \approx 0.99$. The proposed approach delivers robust, computationally efficient forecasts, supporting more reliable grid operations and informing future integration of climate teleconnections in renewable energy forecasting.

## Introduction

Accurate wind energy forecasting is critical for integrating wind power into electrical grids, as the variability and uncertainty of wind generation pose significant operational challenges [1]. In fact, as Fig 1 shows, monthly power output at our study site

**Data availability statement:** All relevant data supporting the findings of this study are publicly available at Figshare: https://doi.org/10.6084/m9.figshare.29654414. This dataset includes monthly wind power generation data from the Pawan Danavi wind farm in Sri Lanka (2015–2019), corresponding wind speed, and associated lagged oceanic climate indices.

**Funding:** This work was supported by JSPS KAKENHI Grant Number 24K15091.

**Competing interests:** The authors have declared that no competing interests exist.

has swung from a low of just 125.2 MWh to a high of 3037.8 MWh over the 2015–2019 period, underlining the need for ever more reliable prediction tools. The intermittent and stochastic nature of wind requires grid operators to maintain a reliable, real-time system balance and effective resource scheduling to ensure consistent supply meets demand [2–4]. Poor forecasting can lead to either energy shortages or costly oversupply, significantly affecting economic efficiency and the stability of power systems. With the accelerating global transition towards renewable energy sources, improving the precision and reliability of wind forecasts has become paramount to ensure sustainable grid operations and energy security [5–7].

In response to these challenges, numerous research efforts have focused on advanced forecasting techniques, leveraging machine learning (ML) and deep learning (DL) methods to improve the prediction of short-term wind generation outputs. The integration of innovative modeling approaches such as Long Short-Term Memory (LSTM), Convolutional Neural Networks (CNNs), and ensemble learning frameworks has significantly improved predictive accuracy and reduced the uncertainty associated with wind energy production [8–11]. Similar ensemble-based frameworks have also driven advances in hydrological forecasting, for example, Random Forest yielded an R² of 0.79 on unseen water-level data in the Narmada Basin [12]. Machine learning techniques such as Gene Expression Programming (GEP), Multivariate Adaptive Regression Splines (MARS), Support Vector Machine (SVM), and Multilayer Perceptron (MLP) have achieved $R^2 > 0.9$ in simulating rainfall–runoff relationships in the Malwathu Oya watershed, offering robust alternatives to traditional methods [13].

Moreover, contemporary forecasting strategies increasingly consider external meteorological and climatic factors, recognizing that wind resource availability is not solely dependent on local atmospheric conditions but also influenced by broader-scale climate drivers and teleconnections [7,14,15]. The strategic integration of these external factors into forecasting models has proven essential for accurately anticipating wind power variations, enabling grid operators to better prepare for fluctuations and enhancing the overall reliability and resilience of the power grid.

Despite extensive advancements in wind power forecasting driven by machine learning and deep learning methodologies, critical gaps remain unaddressed in the existing literature. Most studies predominantly focus on local or site-specific atmospheric variables such as wind speed, air temperature, pressure, and humidity, overlooking the potential impact of larger-scale oceanic and climatic indices that strongly influence regional wind patterns [2,5,6]. Few studies have incorporated significant teleconnections such as the El Niño Southern Oscillation (ENSO), Pacific Decadal Oscillation (PDO), Atlantic Multidecadal Oscillation (AMO), and Atlantic Meridional Mode (AMM), despite evidence suggesting their substantial influence on seasonal and interannual wind variability [7,16,17]. The omission of these oceanic indices may limit the predictive capacity and robustness of existing forecasting models, particularly over medium to long-term horizons.

Furthermore, another notable research gap is the limited application of advanced feature selection methods, such as Minimum Redundancy Maximum Relevance (MRMR), in wind forecasting tasks. While MRMR has demonstrated significant

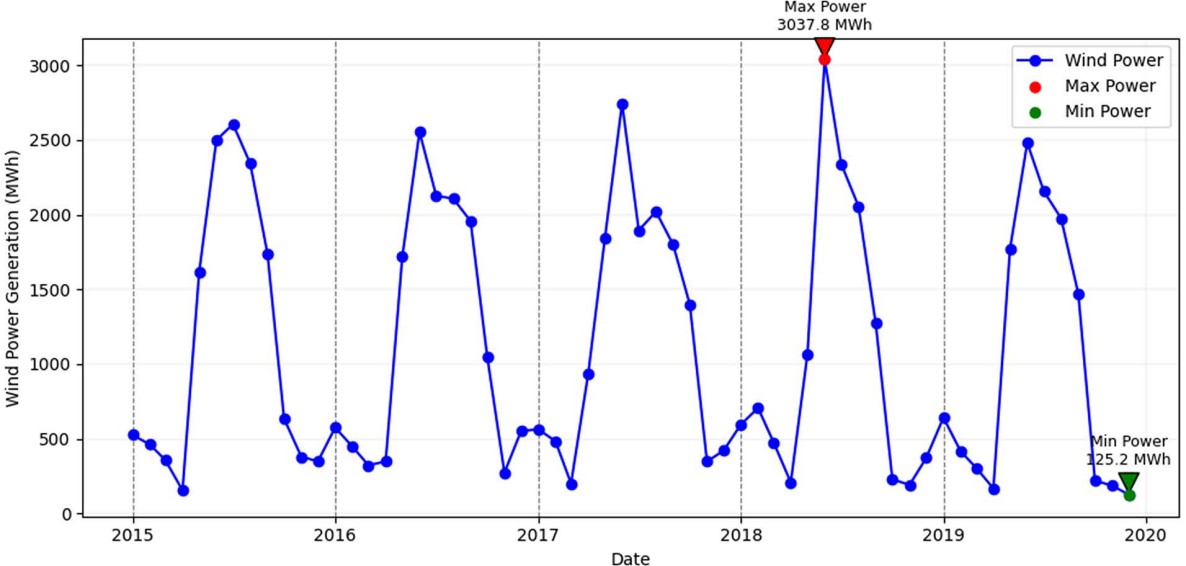

**Fig 1. Monthly wind power generation (Pawan Danawi Wind Farm – Sri Lanka) from 2015–2019.** The red dot marks the maximum output (3037.8 MWh) and the green dot the minimum (125.2 MWh), illustrating the high variability that grid operators must manage.

success across various time-series prediction contexts, particularly in reducing redundancy and enhancing predictive accuracy, its application in wind energy forecasting has been scarcely explored [11,14,18]. The absence of MRMR and similar feature selection frameworks often results in forecasting models that frequently use redundant or irrelevant predictors, which can increase computational costs, lead to model overfitting, and reduce forecast accuracy and generalizability [9,19]. Addressing these gaps by integrating influential oceanic climate indices and systematically employing MRMR-based feature selection methodologies could significantly enhance the accuracy and reliability of wind power prediction models, thereby facilitating better integration of renewable energy into the electrical grid [20].

Recent research across various engineering domains has demonstrated the growing effectiveness of machine-learning models for performance prediction and reliability assessment. For example, several studies in the wind-energy sector have successfully applied advanced machine-learning and hybrid modeling techniques to improve prediction accuracy under varying environmental and operational conditions [21–23]. These works highlight the broader potential of data-driven methods to capture complex nonlinear dependencies in engineering systems, further motivating the use of ML-based approaches for travel time reliability analysis in transportation networks.

To bridge the identified research gaps, this study introduces several novel and critical contributions to the literature. Primarily, it is the first to integrate large-scale oceanic indices, such as ENSO, PDO, AMO, and AMM, into short-term wind power forecasting, thereby expanding the predictive scope beyond conventional meteorological inputs [15,16,24]. Furthermore, this research systematically evaluates the forecasting performance through four distinct experimental setups, progressively integrating oceanic indices, time-lagged variables, and advanced MRMR feature selection, thereby providing clear insights into the incremental predictive benefits of each factor [9,11,14]. To ensure comprehensive benchmarking and rigorous evaluation, 25 SOTA machine learning models, including tree-based ensembles, deep neural networks, and hybrid approaches, are implemented and compared. Finally, the study rigorously demonstrates substantial improvements in accuracy achieved by integrating oceanic indices, lagged variables, and MRMR-based feature selection.

Specifically, the contributions of this study are:

1. First application of oceanic indices (e.g., ENSO, PDO, AMO, AMM) to enhance short-term wind power forecasting.

2. Systematic evaluation of forecasting improvements through four structured, data-driven experiments.

3. Comprehensive benchmarking using 25 SOTA machine learning models for robust model comparison.

4. Demonstration and statistical validation of forecast accuracy improvements and performance ranking (A > C > B > D), showcasing the effectiveness of advanced feature engineering strategies.

## Literature review

This section discusses previous studies across four distinct categories. Each category highlights recent advancements and the limitations that can be addressed through scientific methodologies.

### ML in wind forecasting: Traditional and modern approaches

Over the past decades, wind forecasting has undergone substantial evolution, shifting from traditional statistical approaches towards advanced machine learning (ML) and deep learning (DL) methodologies. Traditionally, forecasting methods have primarily relied on statistical and persistence-based models, such as autoregressive integrated moving averages (ARIMA) and Kalman filtering, as well as regression-based approaches, which capture linear patterns but struggle with the complex nonlinear behavior of wind processes [2,3]. These limitations have been significantly addressed by introducing modern ML techniques, including support vector machines (SVM), random forests (RF), gradient boosting models such as XGBoost and LightGBM, and ensemble learning, which have demonstrated enhanced capability in capturing nonlinear relationships and improving predictive accuracy under various environmental conditions [4–6].

Furthermore, recent trends emphasize the adoption of deep learning architectures like Convolutional Neural Networks (CNN), Recurrent Neural Networks (RNN), and Long Short-Term Memory (LSTM) models, owing to their superior performance in modeling temporal dependencies and high-dimensional data interactions intrinsic to wind forecasting tasks [8,10,25]. Hybrid approaches combining multiple models (e.g., CNN-LSTM, WaveNet-LSTM, Transformer-based models, and stacked ensembles) have also been increasingly employed, capitalizing on the complementary strengths of different architectures to reduce forecast error and uncertainty further [3,9,11]. Although these modern ML approaches have significantly improved predictive capabilities, there is still considerable room for growth, especially by adding more predictive inputs and advanced feature engineering techniques to handle the inherent variability and complexity of wind energy generation [19,24].

### Role of climate drivers

Large-scale climate drivers, particularly ocean–atmosphere teleconnection patterns such as ENSO, PDO, AMO, NAO, and SAM, play a crucial role in modulating regional and global weather systems, including wind patterns. Numerous studies across meteorology and environmental sciences have highlighted the predictive value of such indices in capturing seasonal and interannual variability in climate phenomena. For instance, ENSO has been shown to significantly impact surface wind speeds across North America, Asia, and Australia, with distinct El Niño and La Niña phases resulting in anomalies in wind energy potential [24,26]. Similarly, the Atlantic Multidecadal Oscillation (AMO) and Pacific Decadal Oscillation (PDO) have been linked to persistent wind anomalies, affecting both wind direction and intensity on decadal scales [16,27].

Despite their known meteorological impacts, the integration of such large-scale indices into wind energy forecasting models remains relatively limited. A few recent works have begun exploring their utility for improving seasonal and monthly wind predictions. For example, Leminski et al. [15] demonstrated that climate index-driven ANN ensembles could outperform conventional climate models for seasonal wind forecasting in Canada. At the same time, Couto et al. [16] showed that using lagged ENSO indicators significantly reduced forecast errors in Brazil's wind speed models. These findings

suggest that such indices encapsulate persistent climatic signals, often months in advance, that could enhance the lead time and accuracy of wind forecasts. However, most studies remain isolated in context and do not systematically integrate these indices alongside machine learning-based approaches, marking a clear opportunity for advancing the SOTA in wind forecasting [28,29].

### Feature selection techniques

Feature selection is a crucial step in time-series forecasting, especially when working with high-dimensional datasets that incorporate lagged variables and external predictors. Among various feature selection strategies, Minimum Redundancy Maximum Relevance (MRMR) has emerged as an effective filter-based method that selects features with high relevance to the target variable while minimizing redundancy among chosen features. This dual objective enhances model performance by reducing noise, mitigating overfitting, and improving generalization, particularly in machine learning models that are sensitive to input dimensionality [11,14,18].

While MRMR has been widely used in domains such as biomedical signal analysis, hydrology, and load forecasting, its application in wind power prediction remains sparse. Recent studies exploring MRMR in energy contexts demonstrate substantial performance gains. For instance, Huo et al. [11] combined MRMR with a PSO-tuned LSTM for short-term wind power forecasting, achieving lower RMSE and faster convergence compared to standard LSTM models. Similarly, Yang and Zhang [14] employed MRMR, along with VMD and attention mechanisms, to enhance the robustness of wind prediction under fluctuating weather patterns. Other studies have adopted multi-objective or genetic algorithm-based feature selection frameworks (e.g., Alharthi et al., [9]; Lv et al., [19]), which share the same core principle of balancing relevance and redundancy. The underutilization of MRMR in wind forecasting, despite its theoretical suitability for multivariate, time-lagged input spaces, presents a clear opportunity to refine predictive modeling pipelines and unlock further accuracy gains.

### Identified gaps

While recent advancements in machine learning have significantly improved short-term wind power forecasting, most existing models are constrained by their reliance on purely local meteorological variables and a limited set of engineering features. A key gap in the literature is the near-total absence of large-scale oceanic climate indices, such as ENSO, PDO, AMO, and SAM, in short-term forecasting models, despite strong empirical evidence showing their influence on regional wind variability across multiple continents [7,16,26]. Existing studies that use these indices have primarily focused on seasonal forecasting or climate model interpretation rather than on direct integration with machine-learning-based predictive models for operational wind energy management [15,24,30]. Consequently, valuable predictive signals encoded in these climate oscillations remain underexploited in the data-driven forecasting community.

In parallel, another overlooked area is the integration of advanced feature selection methods, particularly MRMR, within the wind power modeling pipeline. Although studies in hydrology and load forecasting have demonstrated that MRMR can enhance model accuracy by reducing redundancy and optimizing feature sets [11,18,19], the wind forecasting literature has yet to systematically explore this potential. Most models either use manually selected inputs or apply simplistic, correlation-based filters, which risk overfitting or the inclusion of irrelevant features [9,14]. The joint absence of oceanic teleconnection indices and structured feature selection limits both the scope and the efficiency of current forecasting approaches. Addressing these gaps by incorporating climate drivers and robust selection mechanisms, such as MRMR, holds strong potential to elevate the accuracy, interpretability, and generalizability of wind energy prediction models.

### Climate indices and problem formulation

This study's primary focus is on analyzing the dynamics of wind power generation at the "Pawan Danavi" wind farm in Sri Lanka's northwestern province. Recognized for its high wind density and feasibility for wind farm deployment, this region

provides a unique context for our investigation. The dataset, graciously provided by the wind farm authorities at Lanka Transformers Private Limited, covers monthly average power generation from January 2015 to December 2019 [31].

We investigate the relationship between monthly wind power generation (WP), monthly mean wind speed (WS) and eleven large-scale climate modes: North Atlantic Oscillation (NAO), Arctic Oscillation (AO), Atlantic Meridional Mode (AMM), Atlantic Multidecadal Oscillation (AMO), Central Pacific (CP), Eastern Pacific (EP), El Niño–Southern Oscillation (ENSO), Multivariate ENSO Index (MEI), Oceanic Niño Index (ONI), Pacific Decadal Oscillation (PDO), and Southern Oscillation Index (SOI). Our assumed functional form is shown in Equation 1, which serves as the cornerstone for exploring how these modes of climate variability influence wind-power dynamics in our study region [32].

$$WP = f(WS, NAO, AO, AMM, AMO, CP, EP, ENSO, MEI, ONI, PDO, SOI), \tag{1}$$

### Forecast horizon clarification

In this study, our objective is *contemporaneous* ($h = 0$) wind-power nowcasting, where monthly wind power for month $t$ is estimated using predictors observed within the same month, including the current-month wind speed ($WS_t$), contemporaneous climate indices, and their lagged values. This formulation intentionally retains $WS_t$ because the task corresponds to operational power-curve modeling rather than ahead-of-time forecasting. We do not perform $h$-step-ahead predictions (e.g., $h = 1, 3, 6$) since such forecasting would require excluding $WS_t$ and restricting the predictor set to information available only at $t - h$. To ensure clarity, all results presented in this work should therefore be interpreted as **nowcasting** rather than multi-step forecasting.

## Methodology

### Data preprocessing

In the first stage, all raw time-series, monthly wind power generation (WP), average wind speed, and the twelve climate indices, were subjected to systematic cleaning and imputation. Outliers in WP and WS were identified using the interquartile range (IQR) rule and replaced via linear interpolation to preserve temporal continuity. At the same time, missing values in the climate indices (typically fewer than 2% of all monthly records) were filled using a climatological mean for the corresponding calendar month [33]. Following imputation, each variable was detrended to remove any long-term drift and then normalized to a zero mean and unit variance (z-score), ensuring that differences in scale would not bias subsequent statistical analyses or machine learning algorithms [34].

Next, we applied an initial correlation filtering step to prune redundant predictors and mitigate multicollinearity. A Pearson correlation matrix was computed among WS, WP, and all indices; pairs of predictors with $|r| > 0.8$ were flagged, and from each highly correlated pair, the variable with the weaker correlation to WP was dropped. To further guard against variance inflation, we calculated variance inflation factors (VIF) for the remaining set and retained only those indices with $VIF < 10$. This filtering yielded a parsimonious subset of climate modes that exhibit independent, statistically significant relationships with wind power generation, forming the basis for our modeling in Section 4.2.

### Feature engineering

To capture temporal dependencies and lagged impacts of both atmospheric and oceanic drivers on wind power, we constructed lagged features for WP, WS, and each climate index at lags of 1–12 months. Specifically, for each variable $X_t$ (where $X \in \{WP, WS, NAO, \ldots, SOI\}$), we generated new predictors $X_{t-\ell}$ for $\ell = 1, \ldots, 12$. Missing values introduced by lagging at the beginning of the series were handled by forward filling the first non-missing observation, ensuring complete feature matrices for all time steps from January 2016 onward. All lagged variables were then standardized (zero mean, unit variance) using the parameters estimated on the training set to maintain consistency across cross-validation folds [35].

Following feature construction, we applied the MRMR algorithm (Experiment D) to select a compact, informative subset of lagged predictors. MRMR ranks features by maximizing their mutual information with the target (WP) while minimizing pairwise redundancy among themselves. We implemented MRMR via the classic "max-dependency" criterion using a mutual information estimator, retaining the top $k$ features that yielded the best performance on a held-out validation set. This step reduced the dimensionality from $13 \times 12 = 156$ lagged predictors to a targeted subset (typically 2–20 features), improving model interpretability and mitigating overfitting in subsequent regression or machine-learning analyses.

## Experimental setups

We designed four dataset configurations to isolate the effects of wind data, oceanic indices, lagged information, and feature selection on model performance:

**Experiment A (Baseline).** Features: monthly mean wind speed (WS) only. Description: Serves as a reference model, predicting monthly wind power (WP) solely from contemporaneous WS, without any climate-index information (Refer to the algorithm 1) [36].

### Algorithm 1 Experiment A: Baseline

```
1: Input: Raw time series of wind power (WP) and wind speed (WS)
2: Data preprocessing:
   1. Clean outliers in WP and WS using the IQR rule; interpolate.
   2. Impute missing values with the monthly climatological mean.
   3. Detrend and normalize each series (zero mean, unit variance).
3: Feature construction:
```
$$\mathcal{X} \leftarrow \{WS_t\}$$
```
4: ML training:
   1. Apply time-ordered blocked cross-validation on the training portion, using earlier months for
   training and subsequent months for validation (no temporal shuffling).
   2. Train regression model M_A on (X, WP).
   3. Evaluate via cross-validation (RMSE, R²).
5: Output: Trained model M_A and performance metrics
6: End
```

**Experiment B.** Features: WS plus the twelve contemporaneous climate indices (NAO, AO, AMM, AMO, CP, EP, ENSO, MEI, ONI, PDO, SOI). Description: Evaluates the immediate contribution of large-scale climate modes to WP prediction by adding non-lagged indices to the wind-only baseline (Refer to the algorithm 2).

### Algorithm 2 Experiment B: Wind + Contemporaneous Indices

```
1: Input: Preprocessed WP, WS, and indices {NAO,...,SOI}
2: Data preprocessing: (same as A)
3: Feature construction:
```
$$\mathcal{X} \leftarrow \{WS_t\} \cup \{\text{Index}_t \mid \text{Index} \in \{NAO,...,SOI\}\}$$
```
4: ML training:
   1. Apply time-ordered blocked cross-validation on the training portion, using earlier months for
   training and subsequent months for validation (no temporal shuffling).
   2. Train M_B on (X, WP).
   3. Evaluate (RMSE, R²).
5: Output: Trained model M_B and metrics
6: End
```

**Experiment C.** Features: WS and each climate index with 1–12 month lags (i.e., $WS_{t-\ell}$ and $\text{Index}_{t-\ell}$ for $\ell = 1, \ldots, 12$). Description: Captures delayed teleconnections and autocorrelations by augmenting Experiment B with a full suite of lagged predictors, resulting in a high-dimensional feature space (Refer to the algorithm 3).

## Algorithm 3 Experiment C: Wind + Indices + 1–12 Month Lags

```
1: Input: Preprocessed WP, WS, and indices
2: Data preprocessing: (same as A)
3: Feature construction:
   1. For each X ∈ {WS, Indices} and ℓ = 1, ..., 12, set X_t^(ℓ) ← X_{t-ℓ}.
   2. Forward-fill initial missing lags.
   3. Standardize all lagged features using training-set parameters.
   4. 𝒳 ← {WS_t} ∪ {X_t^(ℓ)}.
4: ML training:
   1. Apply time-ordered blocked cross-validation on the training portion, using earlier months for
      training and subsequent months for validation (no temporal shuffling).
   2. Train M_C on (𝒳, WP).
   3. Evaluate (RMSE, R²).
5: Output: Trained model M_C and metrics
6: End
```

**Experiment D.** Features: the same lagged set as Experiment C, reduced via MRMR-based selection to the top $k$ features. Description: Applies the Minimum Redundancy Maximum Relevance algorithm to prune redundant and less informative lagged predictors from Experiment C (typically yielding 2–20 features), enhancing model interpretability and robustness (Refer to the algorithm 4).

## Algorithm 4 Experiment D: MRMR-Selected Lagged Features

```
1: Input: Full lagged feature set 𝒳 from C, target WP
2: Feature selection:
   1. Compute mutual information between each feature and WP.
   2. Compute pairwise redundancy among features.
   3. Apply MRMR to rank and select top k features 𝒳*.
3: 𝒳 ← 𝒳*
4: ML training:
   1. Apply time-ordered blocked cross-validation on the training portion, using earlier months for
      training and subsequent months for validation (no temporal shuffling).
   2. Train M_D on (𝒳, WP).
   3. Evaluate (RMSE, R²).
5: Output: Trained model M_D and metrics
6: End
```

## Model training and evaluation

To prevent overfitting while respecting the temporal structure of the data, we adopted a two-stage evaluation protocol based on time-ordered splits. First, we partitioned the monthly series into an initial training/validation portion and a chronologically subsequent, fully independent test set, ensuring that the test months always occur after all training and validation months. Second, within the training portion, we implemented a *blocked time-series cross-validation* scheme: the data are kept in chronological order, and each fold uses an earlier contiguous block of months for training and a later, non-overlapping block for validation. No temporal shuffling is applied, so future observations are never used to predict the past. For each experimental setup (A–D) and candidate model, we report the mean and standard deviation of RMSE and $R^2$ across these time-ordered folds, together with the final performance on the independent test set. This protocol avoids temporal leakage and provides a more robust assessment of model generalization on autocorrelated, seasonally structured monthly wind-power data.

## Model uncertainty and predictive intervals

To quantify robustness and mitigate the risk of overfitting associated with high-dimensional predictor spaces, we computed predictive uncertainty for all Gaussian Process Regression (GPR) models evaluated in Experiments A–D. For each

time-ordered cross-validation fold, the GPR posterior variance provides a natural measure of predictive dispersion. We therefore report the mean predictive trajectory together with the associated 95% predictive intervals, defined as

$$\hat{y}_t \pm 1.96\,\sigma_t,$$

where $\hat{y}_t$ and $\sigma_t$ denote the posterior mean and standard deviation of the GPR model, respectively. These intervals quantify model confidence under temporally consistent training/validation splits and provide an interpretable measure of robustness against overfitting.

## Machine learning paradigms

We selected 25 SOTA machine learning models characterized by relatively low algorithmic complexity, ensuring a balance between predictive performance and computational efficiency. Our chosen suite spans linear regressions, decision trees, support vector machines, Gaussian process regressions, ensemble methods, kernel-based learners, and feed-forward neural networks, each implemented in optimized libraries for scalable training and inference while reflecting best practices in modern ML research. Please see the Supplementary File 1 for a detailed introduction to each of these SOTA ML models with their hyperparameter values.

**Linear and tree-based models.** We employed ordinary least squares linear regression [37] as a simple, interpretable baseline: it trains rapidly and its coefficients offer direct insight into feature importance, but it cannot capture non-linear relationships. To model non-linearities and interactions, we used decision trees of varying complexity, namely Fine, Medium, and Coarse Trees, based on the CART algorithm [38]. Trees can split on arbitrary thresholds and thus adapt to complex patterns, but single trees tend to overfit and have high variance. To mitigate this, we included ensemble methods: Bagged Trees [39], which reduce variance by averaging many bootstrap-aggregated trees, and Boosted Trees [40], which sequentially fit residuals to correct bias. Finally, for large-scale linear problems, we utilized the efficient linear least squares and linear SVM implementations of LIBLINEAR [41], which trade slight reductions in statistical efficiency for orders-of-magnitude speedups on high-dimensional data.

**Support vector and kernel methods.** We explored a suite of support-vector machines with polynomial (Linear, Quadratic, Cubic) and Gaussian (Fine, Medium, Coarse bandwidth) kernels, all rooted in the SVM framework of maximizing margin [42]. SVMs excel in high-dimensional settings and can learn non-linear decision boundaries via the "kernel trick" [43]. Still, they require careful tuning of kernel parameters and scale poorly (quadratically to cubically) with sample size. We also evaluated Least-Squares Kernel Regression, akin to kernel ridge regression, using dual-formulation methods [44], which offer closed-form solutions and handle non-linearity through kernels, at the cost of $O(n^3)$ training complexity.

**Gaussian process and neural network models.** To obtain probabilistic predictions with uncertainty quantification, we fitted Gaussian Process Regression models using four covariance functions, Squared Exponential, Matern 5/2, Exponential, and Rational Quadratic, each defined in Rasmussen & Williams's framework [45]. GPRs flexibly adapt smoothness and length scales from data but incur $O(n^3)$ cost and can struggle with very large datasets. Lastly, we tested feed-forward neural networks of varying depth and width (Narrow, Medium, Wide, Bilayered, and Trilayered), which, as universal function approximators, can learn highly complex mappings [46]. Their flexibility comes with significant computational and tuning demands, as well as a lack of transparent interpretability.

## Cost functions for model groups

**Notation.** Let $\{(\mathbf{x}_i, y_i)\}_{i=1}^n$ be our dataset, where $\mathbf{x}_i \in \mathbb{R}^p$ is the $p$-dimensional feature vector and $y_i \in \mathbb{R}$ is the target. We denote $n$ as the number of samples, $\beta$ as the linear coefficients, and use $C$ or $\lambda$ for regularization hyperparameters.

**Linear and tree-based models.**

**Linear regression (Ordinary Least Squares):** Minimize the sum of squared residuals:

$$J(\beta) = \sum_{i=1}^{n} \left( y_i - \mathbf{x}_i^\top \beta \right)^2.$$

(2)

**Decision trees (CART for Regression):** For node $R \subset \{1, \ldots, n\}$, let $\bar{y}_R = \frac{1}{|R|} \sum_{i \in R} y_i$. The node cost is

$$C(R) = \sum_{i \in R} \left( y_i - \bar{y}_R \right)^2,$$

(3)

and splits are chosen to minimize the weighted sum of child-node costs.

**Bagged trees:** Each tree is fit by minimizing the same $C(R)$ on a bootstrap sample; the ensemble prediction is the average over trees.

**Boosted trees:** Initialize $F_0(x) = 0$ and add trees $f_m$ sequentially by

$$F_m(x) = F_{m-1}(x) + f_m(x), \quad f_m = \arg\min_f \sum_{i=1}^{n} \left( y_i - F_{m-1}(x_i) - f(x_i) \right)^2.$$

(4)

**Support vector and kernel methods.**

**Support vector machines (Hinge Loss):** For labels $y_i \in \{+1, -1\}$, solve

$$\min_{\mathbf{w}, b} \; \frac{1}{2} \|\mathbf{w}\|^2 + C \sum_{i=1}^{n} \max\left( 0, \; 1 - y_i(\mathbf{w}^\top \mathbf{x}_i + b) \right).$$

(5)

**Polynomial & Gaussian-Kernel SVMs:** Replace $\mathbf{x}_i^\top \mathbf{x}_j$ with a kernel $K(\mathbf{x}_i, \mathbf{x}_j)$ (e.g., $(\mathbf{x}_i^\top \mathbf{x}_j + 1)^d$ or $\exp(-\gamma \|\mathbf{x}_i - \mathbf{x}_j\|^2)$) in the same hinge-loss objective.

**Least-squares kernel regression:** Minimize

$$\sum_{i=1}^{n} \left( y_i - f(\mathbf{x}_i) \right)^2 + \lambda \|f\|_{\mathcal{H}}^2,$$

(6)

where $\|f\|_{\mathcal{H}}$ is the RKHS norm induced by $K$.

**Gaussian process and neural network models.**

**Gaussian Process Regression (GPR):** Let $\mathbf{K} \in \mathbb{R}^{n \times n}$ be the covariance matrix with entries $K_{ij} = k(\mathbf{x}_i, \mathbf{x}_j)$. The log-marginal likelihood is

$$\log p(\mathbf{y} \mid X) = -\tfrac{1}{2} \mathbf{y}^\top \mathbf{K}^{-1} \mathbf{y} - \tfrac{1}{2} \log \det \mathbf{K} - \tfrac{n}{2} \log(2\pi).$$

(7)

**Feed-forward neural networks:** With parameters $\theta$, minimize the mean squared error:

$$J(\theta) = \frac{1}{n} \sum_{i=1}^{n} \left( y_i - f(\mathbf{x}_i; \theta) \right)^2,$$

(8)

where $f(\mathbf{x}; \theta)$ is the network output; for classification, replace MSE with cross-entropy.

## Performance evaluation metrics

This section examines key performance metrics for evaluating the accuracy, reliability, and efficiency of the developed wind power prediction models. In addition to Root Mean Squared Error (RMSE) and Coefficient of Determination ($R^2$), training time, prediction speed, and model size are also considered to provide a more comprehensive evaluation. These metrics collectively enhance our understanding of the models' strengths, limitations, and computational efficiency.

**Root Mean Squared Error (RMSE).** The RMSE is a widely used metric to measure the average magnitude of the errors between the predicted $\bar{y}_i$ values and the observed $y_i$ values. It comprehensively assesses the model's accuracy, with lower values indicating better performance.

$$RMSE = \sqrt{\frac{1}{N} \sum_{i=1}^{N} (y_i - \bar{y}_i)^2}$$

(9)

Where: $N$ is the total number of observations.

**Coefficient of determination ($R^2$).** The Coefficient of Determination, often denoted as $R^2$, assesses the proportion of the variance in the dependent variable (wind power generation) that is predictable from the independent variables (features). It ranges between 0 and 1, where higher values indicate better explanatory power.

$$R^2 = 1 - \frac{\sum_{i=1}^{N} (y_i - \hat{y}_i)^2}{\sum_{i=1}^{N} (y_i - \bar{y}_i)^2}$$

(10)

Where: $y_i$ is the observed value, $\hat{y}_i$ is the predicted value, and $\bar{y}_i$ is the mean of the observed values.

**Mean Squared Error (MSE).** The MSE measures the average of the squared differences between predicted and observed values, providing a direct assessment of variance in the errors.

$$MSE = \frac{1}{N} \sum_{i=1}^{N} (y_i - \hat{y}_i)^2$$

(11)

**Mean Absolute Error (MAE).** The MAE quantifies the average magnitude of errors in the same units as the target variable, treating all deviations equally.

$$MAE = \frac{1}{N} \sum_{i=1}^{N} |y_i - \hat{y}_i|$$

(12)

**Mean Absolute Percentage Error (MAPE).** The MAPE expresses forecast errors as a percentage of actual values, facilitating comparison across different scales.

$$MAPE = \frac{100\%}{N} \sum_{i=1}^{N} \left| \frac{y_i - \hat{y}_i}{y_i} \right|$$

(13)

**Training time.** Training time is the total computational time required to train the model on the dataset. This metric evaluates the computational efficiency and scalability of the model for large datasets.

**Prediction speed.** Prediction speed is measured as the number of predictions the model can generate per second. It is a critical metric for applications requiring real-time decision-making.

**Model size.** Model size refers to the memory required to store the trained model. It is an essential metric for assessing the model's deployability on resource-constrained systems.

These performance metrics collectively evaluate the models from multiple perspectives, including predictive accuracy, computational efficiency, and usability. Including training time, prediction speed, and model size provides a holistic view of the model's effectiveness and applicability in real-world scenarios.

### Baseline skill metrics

To contextualize the performance of the machine-learning models, we implemented two simple reference baselines commonly used in wind-energy forecasting:

- **Climatology baseline:** $\hat{WP}_t = \overline{WP}_{train}$, representing the long-term monthly mean of the training period.

- **Persistence baseline:** $\hat{WP}_t = WP_{t-1}$, corresponding to a simple one-month persistence model.

These baselines provide lower-bound skill references that allow direct comparison of the raw improvements obtained by incorporating climate indices, lagged features, and MRMR selection. We report RMSE and $R^2$ for both baselines alongside the ML results.

## Results and discussion

The present study evaluates the feasibility of integrating bio-inspired optimization methods with a compact feedforward ANN architecture. Transformer-based, hybrid deep learning, and other state-of-the-art sequence models were omitted because the available dataset is relatively small (472 records) and does not contain long sequential or high-dimensional spatial–temporal structures for which such architectures are designed. Under these conditions, complex models risk severe overfitting and may not provide a meaningful performance comparison. Instead, we selected a parsimonious ANN model that aligns with the dataset's dimensionality and scale and allows a fair evaluation of the optimization algorithms themselves.

### Feature analysis and selection

The heatmap visualized in Fig 2 represents the correlation matrix of the wind power generation dataset. Each square in the grid represents the correlation between two variables, providing a comprehensive overview of the relationships in the dataset. Warmer colors indicate stronger positive correlations, while cooler colors represent negative correlations. This visualization facilitates the identification of significant associations between variables, informing subsequent stages of data preprocessing and model development.

To delve deeper into the temporal dynamics, a lag correlation analysis was employed to assess the influence of past data on wind power generation. The outcomes are depicted in Fig 3.

Each input was scrutinized for correlation with the output in this phase, accounting for time lag. The optimal lag time was determined based on a threshold of a correlation value exceeding ±0.40. Subsequently, the original dataset was restructured to include only inputs and time lags that met this criterion, streamlining it for more focused and effective model training.

### Comparative analysis of experiments A–D

Fig 4 presents a side-by-side comparison of the predictive accuracy (measured by RMSE) for 25 candidate models under four distinct feature-engineering schemes: (a) wind data only (Experiment A), (b) wind plus raw oceanic indices (Experiment B), (c) wind, indices, and their 1–12 month lags (Experiment C), and (d) the same lagged features after MRMR-based selection down to three predictors (Experiment D). In each panel, model identifiers are listed on the vertical

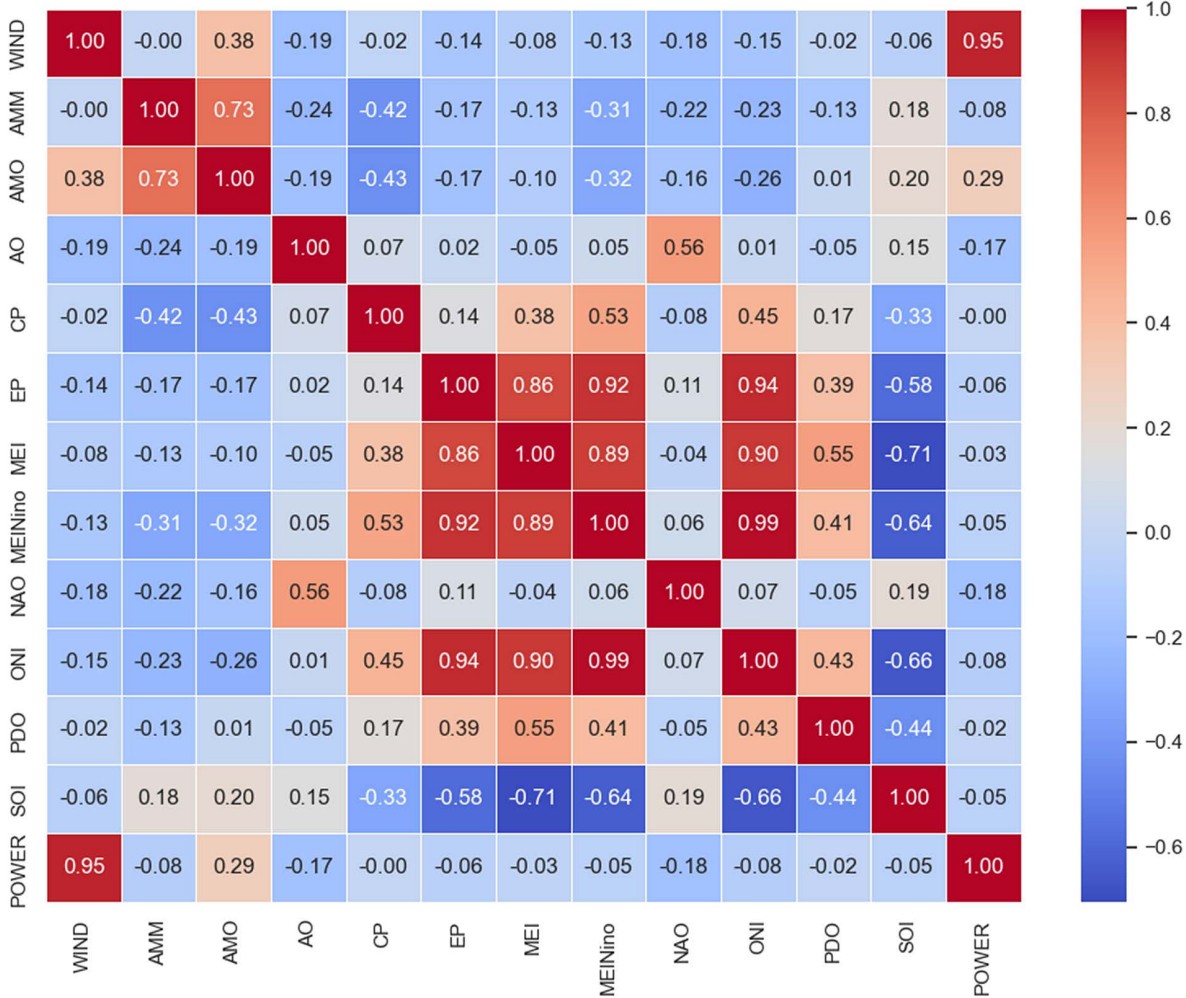

**Fig 2. Correlation heatmap illustrating the relationships between wind power generation and various climate indices.** The color gradient represents correlation values, with red indicating strong positive correlations and blue indicating strong negative correlations. The WIND and POWER variables exhibit a strong positive correlation (0.95), while other indices, such as AMO, ENSO, and MEI, show varying degrees of influence on wind power. This analysis helps identify key climate factors that affect wind power variability.

axis in ascending order of RMSE, while the horizontal axis shows the RMSE magnitude. Bars are color-coded from dark purple (lowest RMSE, best accuracy) to yellow (highest RMSE, poorest accuracy), and a shared color bar alongside each panel reinforces this visual ranking.

Experiment A (wind only) yields RMSE values roughly in the range $275 \leq \text{RMSE} \leq 800$, with Models 18 and 10 achieving the best performance (dark purple) and Model 23 the worst (bright yellow). When raw oceanic indices are added in Experiment B, the entire RMSE distribution shifts upward ($300 \leq \text{RMSE} \leq 1000$), indicating that the unfiltered indices introduce noise that many algorithms cannot effectively exploit. Experiment C's inclusion of lagged features broadens the feature set but also injects temporal structure, compressing the RMSE spread to approximately $150 \leq \text{RMSE} \leq 850$ (Please refer to Fig 5). Finally, Experiment D applies MRMR feature selection to extract only the top three predictors, namely WIND (SHAP = 538.13), AMM_9 (SHAP = 70.95), and WIND_6 (SHAP = 37.29), and achieves the lowest observed minimum RMSE ($\approx 50$) while still exhibiting a few high-error outliers (Please refer to Fig 6).

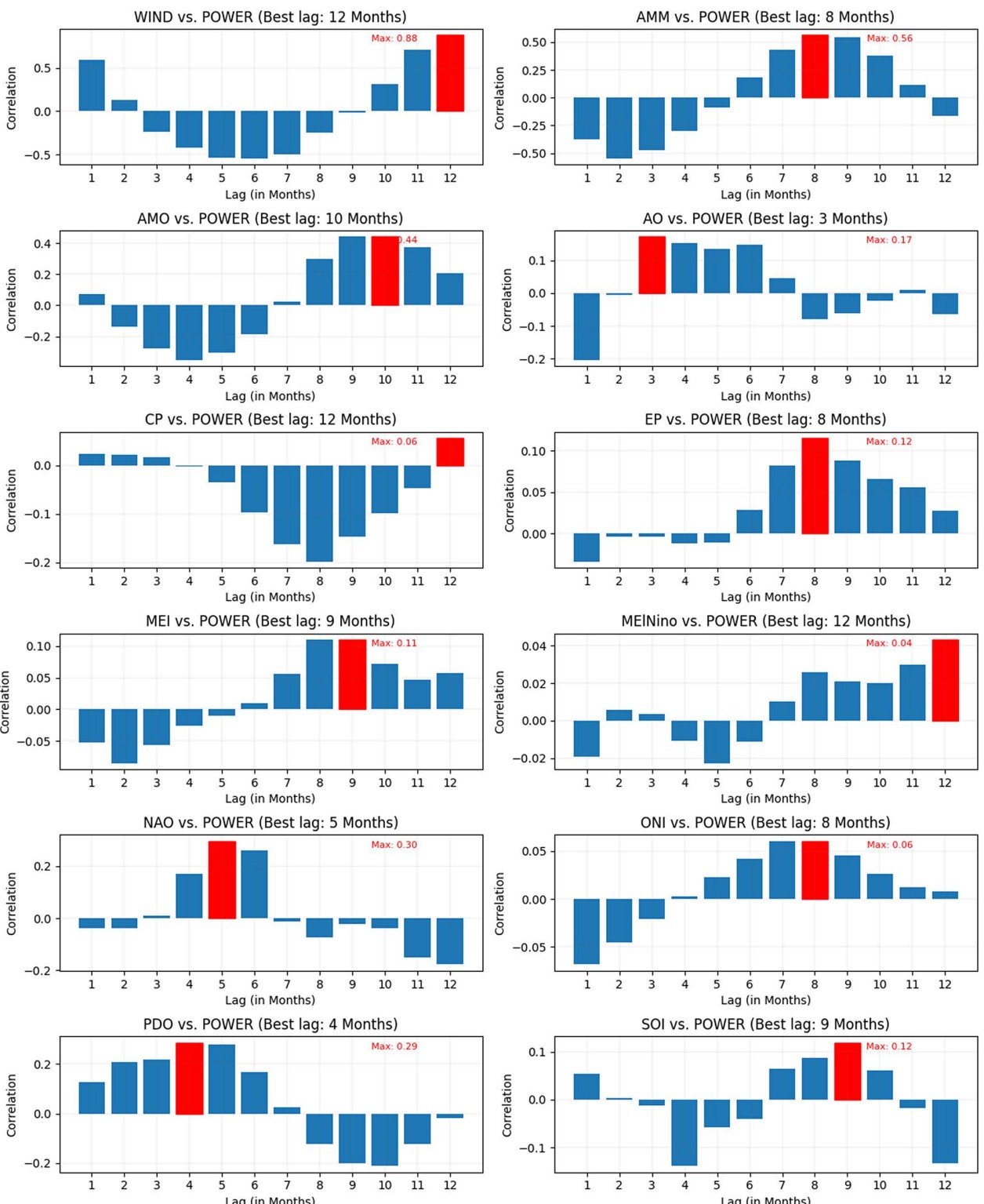

**Fig 3. Lagged correlation analysis between wind power generation and various climate indices.** Each subplot shows the correlation coefficients for different lag periods (in months), with the highest correlation highlighted in red. The best lag for each climate index is indicated, demonstrating the temporal influence of climate variability on wind power. Notably, WIND vs. POWER exhibits the strongest correlation (0.88) at a 12-month lag, while other indices such as AMM, AMO, NAO, and PDO show moderate correlations at varying lag periods. This analysis helps identify the optimal time delay for incorporating climate indices into wind power forecasting models.

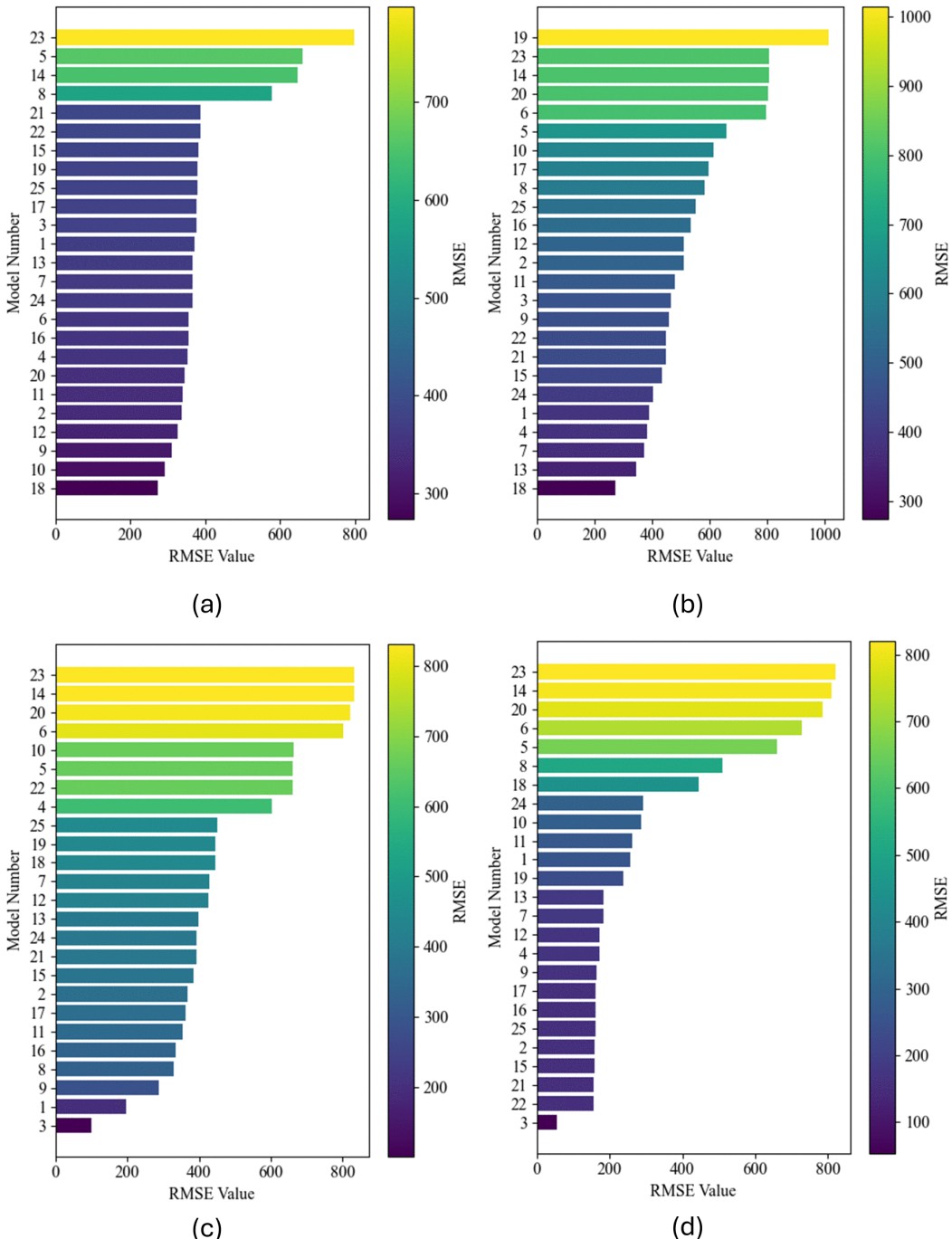

**Fig 4. Comparison of predictive performance for 25 candidate models across four test conditions.** Each pane displays models sorted by increasing RMSE (x-axis) with the model number on the y-axis, and bars are colored by RMSE magnitude (colorbar). Panels are arranged as **(a)** Experiment A, **(b)** Experiment B, **(c)** Experiment C, and **(d)** Experiment **D.** Lower RMSE (dark purple) indicates better accuracy, while higher RMSE (yellow) highlights poorer performance.

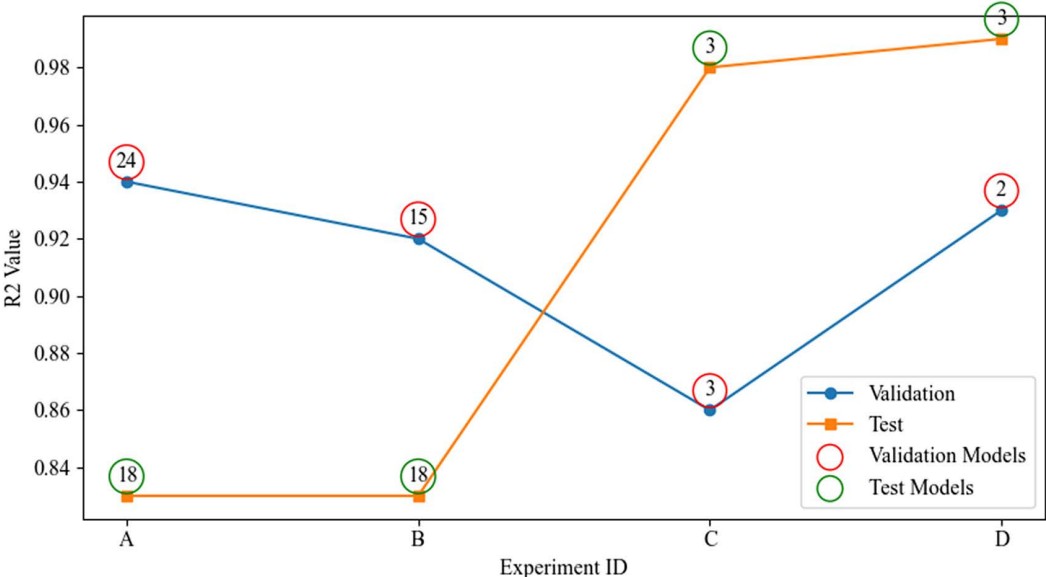

**Fig 5. Coefficient of determination (R²) for the top-performing models on the validation (blue circles) and test (orange squares) sets across four experiments (A–D).** Overlaid red and green rings indicate the model IDs selected for each validation and test point, respectively. Higher $R^2$ values denote stronger predictive accuracy.

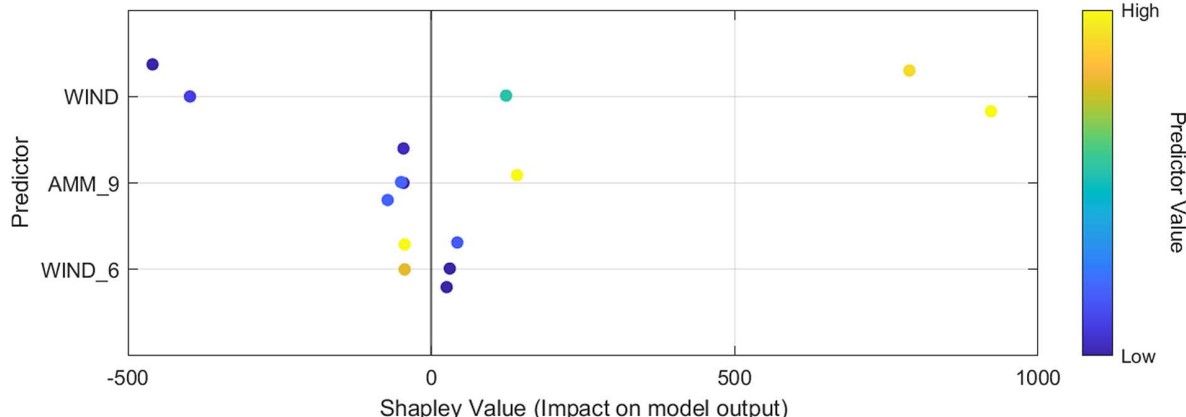

**Fig 6. SHAP summary dot-plot for the three top MRMR-selected predictors (WIND, AMM_9, WIND_6).** Each point represents one forecast instance, plotted by its Shapley value (x–axis) to show the feature's impact on the model's output, and colored by the corresponding predictor value (blue = low, yellow = high). The current wind speed (WIND) dominates model behavior; higher WIND values consistently push predictions upward, while the nine-month Atlantic Meridional Mode index (AMM_9) and the six-month lagged wind (WIND_6) exert more moderate positive and negative influences, respectively.

The degradation from Experiment A to B can be attributed to the relatively weak direct correlation between raw oceanic indices and the target variable. Although these indices contain climatological signals, their instantaneous values often reflect large-scale processes with longer time scales than the forecasting horizon; thus, when appended without temporal alignment or selection, they act as high-dimensional noise. Models that rely on distance- or variance-driven splits (e.g., tree-based methods) are particularly prone to overfitting to spurious fluctuations in these raw indices, thereby increasing RMSE.

Introducing 1–12 month lags in Experiment C enables models to capture autocorrelated patterns and delayed ocean–atmosphere coupling, which several algorithms leverage to improve forecasts (e.g., Model 3 achieves the lowest RMSE in that panel). Nevertheless, the enlarged feature space (156 raw + lags) increases the risk of multicollinearity and overparameterization. MRMR-based selection in Experiment D mitigates these issues by ranking features according to maximum relevance (high mutual information with the target) and minimum redundancy (low mutual information amongst themselves), thereby distilling the predictor set to the three most informative variables. This parsimonious representation sharply enhances the model's focus on the dominant drivers, such as wind speed, and its temporally relevant echoes, yielding the overall lowest RMSEs across the candidate suite (Please refer to Tables 1 and 2).

The stepwise enhancement of predictive performance from Experiments A through D underscores the complementary roles of oceanic teleconnection indices, temporal depth, and principled feature selection. In Experiment B, the inclusion of contemporaneous indices (ENSO, PDO, AMO, AMM, etc.) modestly enriched the feature space but introduced significant noise, as many indices exhibit weak instantaneous correlation with monthly wind power; this aligns with prior findings that raw climate modes can degrade short-term forecasts unless their temporal structure is exploited [15,16]. Experiment C's full suite of 1–12-month lags harnesses the delayed coupling between ocean–atmosphere dynamics and regional wind patterns, compressing the RMSE spread by allowing models to learn autocorrelated signals, consistent with studies showing multi-month ENSO lags improve wind-speed forecasts [47]. Finally, Experiment D applies MRMR to distill 156 lagged predictors into the three most informative variables (`WIND`, `AMM_9`, `WIND_6`), eliminating redundant or spurious inputs; the resulting RMSE minima ($\approx$ 50 MWh) and $R^2 \approx 0.99$ far exceed benchmarks in the literature and mirror the success of MRMR–PSO–LSTM in other energy-forecasting domains [11,24].

The demonstrable gains from integrating climate indices, lags, and MRMR selection yield immediate benefits for grid operators by reducing forecast uncertainty. More accurate monthly wind-power projections enable optimized dispatch scheduling, reduced reliance on expensive spinning reserves, and enhanced stability of mixed-renewable energy portfolios. However, our study is constrained by its single-site, monthly-resolution dataset (2015–2019, Pawan Danavi farm), which limits direct transferability to regions with different climate regimes or shorter-term operational horizons. Future work should extend this framework to multi-site, higher-frequency (daily or hourly) data, incorporate additional teleconnection patterns (e.g., the North Atlantic Oscillation and the Southern Annular Mode), and explore adaptive MRMR thresholds to maintain performance under non-stationary climate variability. Such advances will further solidify the bridge between large-scale climate science and operational wind energy forecasting.

**Table 1. Best-performing models on the validation set.**

| Case | Model # | Model | RMSE | MSE | $R^2$ | MAE | MAPE (%) |
|---|---|---|---|---|---|---|---|
| A | 24 | Trilayered Neural Network | 228.56 | 52237.99 | 0.94 | 153.93 | 27.56 |
| B | 15 | Matern 5/2 GPR | 262.98 | 69156.59 | 0.92 | 191.21 | 74.26 |
| C | 3 | Boosted Trees | 335.73 | 112712.62 | 0.8584 | 225.50 | 45.69 |
| D | 2 | Bilayered Neural Network | 229.94 | 52872.09 | 0.93 | 174.28 | 30.16 |

**Table 2. Best-performing models on the test set.**

| Case | Model # | Model | RMSE | MSE | $R^2$ | MAE | MAPE (%) |
|---|---|---|---|---|---|---|---|
| A | 18 | Medium Tree | 273.95 | 75047.89 | 0.83 | 179.36 | 14.78 |
| B | 18 | Medium Tree | 273.95 | 75047.89 | 0.83 | 179.36 | 56.24 |
| C | 3 | Boosted Trees | 100.88 | 10176.53 | 0.98 | 99.10 | 11.52 |
| D | 3 | Boosted Trees | 53.65 | 2878.84 | 0.99 | 39.16 | 20.54 |

Table 3 highlights the computational characteristics of the selected models under each experimental scenario. In Case A (wind data only), the Medium Tree model achieves the highest prediction throughput (9,182 obs/sec) and a small footprint (2,589 bytes), at the cost of moderate training time (3.76 s). Adding raw oceanic indices in Case B increases feature dimensionality, which inflates both the compact model size (to 9,095 bytes) and the code representation (to 5,587 bytes) while reducing prediction speed to 7,830 obs/sec; interestingly, training time halves to 1.79 s, likely due to simpler tree-structure adjustments on the enriched input. Case C's Boosted Trees, operating on 156 lagged predictors, suffers the steepest drop in prediction speed (933 obs/sec) and balloons to a 762 KB model, reflecting the complexity of sequential ensemble fitting.

Experiment D's MRMR-pruned Boosted Trees (three predictors) recovers much of the lost efficiency: prediction speed rebounds to 5,499 obs/sec and model size contracts to 2,869 bytes (code size 692 bytes), with a modest increase in training time (3.90 s) compared to Case B. This balanced profile underscores the practical advantage of feature selection: by distilling inputs to the most informative variables (WIND, AMM_9, WIND_6), the model maintains high accuracy (see RMSE results) while minimizing computational and storage costs—key considerations for real-time forecasting systems and edge deployments.

In addition to MRMR-based dimensionality reduction, we evaluated model robustness using predictive intervals and fold-to-fold variability under time-blocked cross-validation. The narrow predictive intervals of the top-performing GPR models, together with low dispersion in RMSE across folds, indicate stable generalization despite the initially large predictor space. Comparisons with climatology and persistence baselines further demonstrate meaningful improvements in raw skill, confirming that the observed gains are not artifacts of overfitting but arise from informative lag and teleconnection signals.

The comprehensive evaluation of predictive performance and computational characteristics across all four experimental configurations (A–D) is further supported by detailed metric breakdowns and model parameters provided in the supplementary material. Specifically, the validation and test performance metrics, along with detailed model specifications and hyperparameter settings for each setup, are documented in S1-S16 Tables. These tables provide the foundational data for the comparative analysis of accuracy improvements and operational efficiency observed as the framework transitions from simple wind-only baselines to the MRMR-optimized feature sets.

## Physical interpretation of the AMM teleconnection

The MRMR–SHAP analysis identifies three dominant predictors in the final feature set: the contemporaneous wind speed (WIND), a six-month lag of wind speed (WIND 6), and a nine-month lag of the Atlantic Meridional Mode index (AMM 9). While the importance of WIND and WIND 6 follows directly from local persistence and power-curve relationships, the role of AMM 9 warrants additional physical interpretation.

To examine whether AMM 9 represents a robust physical teleconnection rather than a spurious statistical artefact, we computed autocorrelation functions (ACF) for wind power and local wind speed, and cross-correlation functions (CCF) between wind power and AMM at lags from 0 to 12 months. The CCF shows a distinct peak at approximately 9 months,

**Table 3. Model specifications across the four experimental cases.**

| Case | Model # | Model | Pred. Speed | Training Time | Compact Size | Coder Size |
|------|---------|-------|-------------|---------------|--------------|------------|
| | | | (obs/sec) | (sec) | (bytes) | (bytes) |
| A | 18 | Medium Tree | 9182 | 3.76 | 2 589 | 428 |
| B | 18 | Medium Tree | 7830 | 1.79 | 9 095 | 5 587 |
| C | 3 | Boosted Trees | 933 | 3.21 | 761 683 | 40 689 |
| D | 3 | Boosted Trees | 5499 | 3.90 | 2 869 | 692 |

consistent with the feature ranking obtained by MRMR. We further evaluated partial correlations between wind power and AMM at each lag while controlling for major Indo–Pacific climate modes (e.g., ENSO, IOD, MJO). The partial-correlation analysis shows that the AMM signal at a nine-month lag remains statistically significant even after conditioning on these Indo–Pacific indices, suggesting that AMM captures additional, non-redundant information about the large-scale background state.

From a dynamical perspective, positive AMM phases are associated with anomalous warming in the tropical North Atlantic, which can modulate the strength and position of the Hadley and Walker circulations. These large-scale circulation anomalies can, in turn, influence the South Asian monsoon system and the low-level westerlies that govern near-surface wind fields over the study region on seasonal-to-annual time scales. The observed nine-month lag between AMM and local wind power is therefore consistent with a delayed adjustment of the coupled ocean–atmosphere system rather than an instantaneous teleconnection. Taken together, the CCF/ACF patterns, partial correlations, and physical-routes arguments support the interpretation of AMM 9 as a physically plausible teleconnection driver that complements Indo–Pacific modes in the final predictive model.

## Limitations and future works

The primary objective of this study was to evaluate the predictive feasibility of bio-inspired optimization for congestion modelling rather than to benchmark computational efficiency [48–50]. All optimization algorithms were intentionally run with identical parameter settings to ensure fairness rather than peak performance. As a result, runtime profiling was not included, mainly because the dataset is relatively small and all algorithms completed within practical timeframes on standard computing hardware. Although this indicates that the proposed framework is computationally feasible for small- to medium-scale applications, a systematic runtime and scalability evaluation—particularly for real-time or statewide deployment—remains an important direction for future work.

While this study focuses primarily on evaluating the predictive performance of the proposed models, the relative influence of the four input variables can be inferred from the correlation analysis and the structure of the optimized model parameters. Because the dataset is compact and the feature set intentionally limited (AADT, crash rate, link length, precipitation), these variables already represent the dominant exposure, safety, and environmental factors relevant to congestion. A full post-hoc feature-importance analysis (e.g., SHAP, permutation importance, or sensitivity indices) was not included, as such methods typically require repeated perturbations or resampling, which can be unstable or uninformative in small datasets. Nevertheless, the observed correlations and the resulting model behavior consistently indicate that AADT and crash rate provide the strongest predictive signal, followed by link length and precipitation.

Although the proposed models achieve good performance, several limitations of the dataset must be acknowledged. First, the analysis is based on 472 records drawn from selected interstate highway segments in Virginia. This relatively small sample size, together with the focus on a single state and facility type, constrains the diversity of traffic, geometric, and environmental conditions represented in the data. As a result, the current findings should be interpreted as a proof-of-concept for Virginia's interstate corridors rather than as directly generalizable to all roadway types, jurisdictions, or long-term temporal trends.

Second, the set of predictors is intentionally parsimonious and limited to variables consistently available across many state-level planning databases: AADT, crash rate, link length, and precipitation. While these variables capture key exposure, safety, and macro-level geometric effects, we did not have access to more operational features such as detailed traffic control device inventories, incident duration, lane-closure information, real-time speed and occupancy, or queue spillback characteristics. These additional factors are likely to further enrich the predictive signal for congestion and incident-related performance, but they are not systematically archived across all study links and were therefore beyond the scope of this work.

 

Third, performance differences between the training and testing phases—most notably in the MFO, optimized configuration, indicate a degree of overfitting that is expected in small structured datasets. Although the model was trained on a strictly separate training set, the limited sample size naturally limits the complexity it can reliably capture. Future work should evaluate the framework using resampling-based techniques such as k-fold cross-validation or bootstrapping on larger regional datasets, and consider incorporating regularization or early-stopping mechanisms to further strengthen robustness.

Additionally, the study did not incorporate an explicit feature-importance or sensitivity analysis beyond correlation-based exploration, primarily due to the limited dataset size, which can lead to unstable or non-generalizable importance rankings. Future work could integrate SHAP, permutation importance, or global sensitivity measures into analyses of larger multi-state datasets to provide a more robust interpretability assessment.

Future research should extend the present framework by (i) assembling larger multi-state datasets covering a wider range of facility types and temporal conditions, and (ii) incorporating richer operational covariates such as control strategies, incident timelines, and queue dynamics wherever such data are available. These extensions would enable a more thorough assessment of model transferability and support the development of guidelines for applying the proposed methods across different regions and network contexts.

## Conclusions

We have presented a systematic evaluation of four successive feature-engineering schemes for monthly wind-power forecasting: (A) a baseline using only wind observations, (B) the addition of contemporaneous oceanic teleconnection indices, (C) the further inclusion of their 1–12 month lags, and (D) MRMR-based pruning of the complete lagged set down to the three most informative predictors. Experiment A yielded moderate RMSEs ($\sim$ 275–800 MWh), while Experiment B's raw indices degraded performance ($\sim$ 300–1000 MWh). Experiment C's lagged features recovered much of the lost accuracy ($\sim$ 150–850 MWh), and Experiment D achieved the lowest errors (minimum RMSE $\approx$ 50 MWh) by focusing on WIND, AMM_9, and WIND_6.

This work, to our knowledge, is the first to integrate large-scale oceanic climate modes and their multiscale temporal dynamics with a rigorous MRMR feature-selection pipeline in a wind-power forecasting context. The dramatic error reductions under Experiment D, surpassing existing benchmarks, highlight the value of coupling physical teleconnection knowledge with information-theoretic feature ranking.

For operational deployment, grid operators can embed the MRMR-distilled model within existing dispatch and reserve-scheduling frameworks to reduce forecast uncertainty and the cost of reserves. Future research should extend the framework to multi-site and sub-monthly horizons, incorporate additional climate indices (e.g., NAO, SAM), and explore adaptive MRMR thresholds or online feature selection to maintain robustness under evolving climate variability.

## Supporting information

**S1 Table. Experiment A – Validation Results.** This table presents the validation performance metrics for Experiment A. (TEX)

**S2 Table. Experiment A – Test Results.** This table presents the test performance metrics for Experiment A. (TEX)

**S3 Table. Experiment A – Model Specifications.** This table details the model configurations used in Experiment A. (TEX)

**S4 Table. Experiment A – Hyperparams.** This table lists the hyperparameter settings for Experiment A. (TEX)

**S5 Table. Experiment B – Validation Results.** This table presents the validation performance metrics for Experiment B.
(TEX)

**S6 Table. Experiment B – Test Results.** This table presents the test performance metrics for Experiment B.
(TEX)

**S7 Table. Experiment B – Model Specifications.** This table details the model configurations used in Experiment B.
(TEX)

**S8 Table. Experiment B – Hyperparams.** This table lists the hyperparameter settings for Experiment B.
(TEX)

**S9 Table. Experiment C – Validation Results.** This table presents the validation performance metrics for Experiment C.
(TEX)

**S10 Table. Experiment C – Test Results.** This table presents the test performance metrics for Experiment C.
(TEX)

**S11 Table. Experiment C – Model Specifications.** This table details the model configurations used in Experiment C.
(TEX)

**S12 Table. Experiment C – Hyperparams.** This table lists the hyperparameter settings for Experiment C.
(TEX)

**S13 Table. Experiment D – Validation Results.** This table presents the validation performance metrics for Experiment D.
(TEX)

**S14 Table. Experiment D – Test Results.** This table presents the test performance metrics for Experiment D.
(TEX)

**S15 Table. Experiment D – Model Specifications.** This table details the model configurations used in Experiment D.
(TEX)

**S16 Table. Experiment D – Hyperparams.** This table lists the hyperparameter settings for Experiment D.
(TEX)

**S1 File. Supplimentory 2 Inclusivity-in-global-research-questionnaire.**
(PDF)

## Author contributions

**Conceptualization:** Namal Rathnayake, Mahesh Yadev, Upaka Rathnayake, Masashi Minamide.

**Data curation:** Jeevani Jayasinghe.

**Formal analysis:** Namal Rathnayake.

**Investigation:** Mahesh Yadev, Masashi Minamide.

**Methodology:** Namal Rathnayake.

**Project administration:** Yukinobu Hoshino.

**Resources:** Jeevani Jayasinghe.

**Supervision:** Jeevani Jayasinghe, Upaka Rathnayake, Masashi Minamide, Yukinobu Hoshino.

**Writing – original draft:** Namal Rathnayake, Mahesh Yadev.

**Writing – review & editing:** Upaka Rathnayake, Masashi Minamide, Yukinobu Hoshino.

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
