## [Decision Letter · Decision Letter 0]

18 Jul 2025

PONE-D-25-17132Wind Power Prediction and Enhanced Reliability Through Machine Learning Models and Oceanic-Atmospheric InsightsPLOS ONE

Dear Dr. Rathnayake,

Thank you for submitting your manuscript to PLOS ONE. After careful consideration, we feel that it has merit but does not fully meet PLOS ONE’s publication criteria as it currently stands. Therefore, we invite you to submit a revised version of the manuscript that addresses the points raised during the review process.

The manuscript presents wind power prediction using multiple machine learning models with diverse ML algorithms. Integrating oceanic and atmospheric information with ML is novel, but much information is omitted in the manuscript, which causes confusion and hinders the impact of this study. Reviewers have provided many useful suggestions. Here are several recommendations the editor has:

Need information on the training and testing dataset, for example, the size of each index. What is the key feature that influences wind power and electricity production? Please provide more information.Need solid comparison among all the ML algorithms, please detail the pros and cons for each ML method. Is there any additional improvement you can make to the ML models to make them best suitable for such wind application? The manuscript stopped at presenting the statistics; providing scientific insights will enhance the paper's quality and impact.Please improve the figure quality. Many of them are very blurry.

Please ignore the review #2, who didn't have the full access to your manuscript.

We believe that reviewer #1 used AI-generated comments and editor team has marked it as questionable, although some of them might be useful for authors to improve the paper's quality. Please adapt the comments as you see fit.

We look forward to receiving your revised manuscript.

Kind regards,

Fan Mei

Academic Editor

PLOS ONE

**Journal Requirements:**

1. When submitting your revision, we need you to address these additional requirements. Please ensure that your manuscript meets PLOS ONE's style requirements, including those for file naming. The PLOS ONE style templates can be found at https://journals.plos.org/plosone/s/file?id=wjVg/PLOSOne_formatting_sample_main_body.pdf and https://journals.plos.org/plosone/s/file?id=ba62/PLOSOne_formatting_sample_title_authors_affiliations.pdf 2. Please include a complete copy of PLOS’ questionnaire on inclusivity in global research in your revised manuscript. Our policy for research in this area aims to improve transparency in the reporting of research performed outside of researchers’ own country or community. The policy applies to researchers who have travelled to a different country to conduct research, research with Indigenous populations or their lands, and research on cultural artefacts. The questionnaire can also be requested at the journal’s discretion for any other submissions, even if these conditions are not met.  Please find more information on the policy and a link to download a blank copy of the questionnaire here: https://journals.plos.org/plosone/s/best-practices-in-research-reporting. Please upload a completed version of your questionnaire as Supporting Information when you resubmit your manuscript. 3. Please note that PLOS ONE has specific guidelines on code sharing for submissions in which author-generated code underpins the findings in the manuscript. In these cases, we expect all author-generated code to be made available without restrictions upon publication of the work. Please review our guidelines at https://journals.plos.org/plosone/s/materials-and-software-sharing#loc-sharing-code and ensure that your code is shared in a way that follows best practice and facilitates reproducibility and reuse. 4. Thank you for stating in your Funding Statement: This work was supported by JSPS KAKENHI Grant Number 24K15091  Please provide an amended statement that declares *all* the funding or sources of support (whether external or internal to your organization) received during this study, as detailed online in our guide for authors at http://journals.plos.org/plosone/s/submit-now. Please also include the statement “There was no additional external funding received for this study.” in your updated Funding Statement. Please include your amended Funding Statement within your cover letter. We will change the online submission form on your behalf. 5. In the online submission form, you indicated that “The data will be available upon request to the corresponding author”. All PLOS journals now require all data underlying the findings described in their manuscript to be freely available to other researchers, either a. In a public repository, b. Within the manuscript itself, or c. Uploaded as supplementary information.This policy applies to all data except where public deposition would breach compliance with the protocol approved by your research ethics board. If your data cannot be made publicly available for ethical or legal reasons (e.g., public availability would compromise patient privacy), please explain your reasons on resubmission and your exemption request will be escalated for approval. 6. We note that Figure 1 in your submission contain map images which may be copyrighted. All PLOS content is published under the Creative Commons Attribution License (CC BY 4.0), which means that the manuscript, images, and Supporting Information files will be freely available online, and any third party is permitted to access, download, copy, distribute, and use these materials in any way, even commercially, with proper attribution. For these reasons, we cannot publish previously copyrighted maps or satellite images created using proprietary data, such as Google software (Google Maps, Street View, and Earth). For more information, see our copyright guidelines: http://journals.plos.org/plosone/s/licenses-and-copyright. We require you to either present written permission from the copyright holder to publish these figures specifically under the CC BY 4.0 license, or remove the figures from your submission: a. You may seek permission from the original copyright holder of Figure 1 to publish the content specifically under the CC BY 4.0 license.   We recommend that you contact the original copyright holder with the Content Permission Form (http://journals.plos.org/plosone/s/file?id=7c09/content-permission-form.pdf) and the following text:“I request permission for the open-access journal PLOS ONE to publish XXX under the Creative Commons Attribution License (CCAL) CC BY 4.0 (http://creativecommons.org/licenses/by/4.0/). Please be aware that this license allows unrestricted use and distribution, even commercially, by third parties. Please reply and provide explicit written permission to publish XXX under a CC BY license and complete the attached form.” Please upload the completed Content Permission Form or other proof of granted permissions as an "Other" file with your submission. In the figure caption of the copyrighted figure, please include the following text: “Reprinted from [ref] under a CC BY license, with permission from [name of publisher], original copyright [original copyright year].” b. If you are unable to obtain permission from the original copyright holder to publish these figures under the CC BY 4.0 license or if the copyright holder’s requirements are incompatible with the CC BY 4.0 license, please either i) remove the figure or ii) supply a replacement figure that complies with the CC BY 4.0 license. Please check copyright information on all replacement figures and update the figure caption with source information. If applicable, please specify in the figure caption text when a figure is similar but not identical to the original image and is therefore for illustrative purposes only.The following resources for replacing copyrighted map figures may be helpful: USGS National Map Viewer (public domain): http://viewer.nationalmap.gov/viewer/The Gateway to Astronaut Photography of Earth (public domain): http://eol.jsc.nasa.gov/sseop/clickmap/Maps at the CIA (public domain): https://www.cia.gov/library/publications/the-world-factbook/index.html and https://www.cia.gov/library/publications/cia-maps-publications/index.htmlNASA Earth Observatory (public domain): http://earthobservatory.nasa.gov/Landsat:
http://landsat.visibleearth.nasa.gov/USGS EROS (Earth Resources Observatory and Science (EROS) Center) (public domain): http://eros.usgs.gov/#Natural Earth (public domain): http://www.naturalearthdata.com/ 7. If the reviewer comments include a recommendation to cite specific previously published works, please review and evaluate these publications to determine whether they are relevant and should be cited. There is no requirement to cite these works unless the editor has indicated otherwise.

Reviewers' comments:

Reviewer's Responses to Questions

**Comments to the Author**

1. Is the manuscript technically sound, and do the data support the conclusions?

Reviewer #1: Yes

Reviewer #2: No

Reviewer #3: Yes

2. Has the statistical analysis been performed appropriately and rigorously? 

Reviewer #1: Yes

Reviewer #2: I Don't Know

Reviewer #3: Yes

3. Have the authors made all data underlying the findings in their manuscript fully available?

Reviewer #1: Yes

Reviewer #2: No

Reviewer #3: Yes

4. Is the manuscript presented in an intelligible fashion and written in standard English?

Reviewer #1: Yes

Reviewer #2: No

Reviewer #3: Yes

5. Review Comments to the Author

**Reviewer #1:** 1. Overview

Summary of Contributions:

This paper explores the use of machine learning models for wind power forecasting, integrating oceanic and atmospheric variables to enhance prediction accuracy. The study employs a diverse set of machine learning algorithms, including XGBoost, Random Forest, LSTM, and ANFIS, to predict wind power generation at a Sri Lankan wind farm. The paper demonstrates that by including climate indices such as ENSO, AMM, and PDO, the predictive performance of the models improves significantly, with the best model achieving an R² of up to 0.97. The work underscores the importance of combining data-driven modeling with environmental variability to enhance wind energy predictability and reliability.

Significance of the Study:

This research is highly significant for the renewable energy sector, specifically in improving the accuracy of wind power predictions. Wind power generation is subject to high variability, and reliable forecasting is crucial for grid integration and efficient energy management. The integration of oceanic and atmospheric indices, alongside traditional machine learning techniques, provides a unique contribution to the field, helping address the uncertainty in renewable energy forecasting.

2. Content and Structure Review

Abstract Evaluation:

The abstract provides a clear summary of the objectives, methodology, and key findings. It effectively communicates the novelty of combining oceanic and atmospheric data with machine learning models. However, it could benefit from a more detailed mention of the practical implications of this research, such as how these predictions could impact energy management or policy decisions related to wind power.

Introduction Evaluation:

The introduction effectively sets the stage by discussing the challenges in wind power forecasting and the potential for machine learning models to improve prediction accuracy. The paper introduces oceanic and atmospheric variability as key factors influencing wind power generation, but a more detailed explanation of how these factors are specifically integrated into the models could help clarify the novelty of the approach.

Conclusion Evaluation:

The conclusion summarizes the key findings and emphasizes the importance of incorporating environmental variability into wind power forecasting. However, it could be expanded to discuss the broader implications of these findings, such as how this approach could be applied to other renewable energy sources or how it might be adapted for real-time forecasting applications.

Suggestions for Improvement:

The abstract could be expanded to emphasize the practical implications of the research.

The introduction could better explain how the oceanic and atmospheric indices are incorporated into the machine learning models.

The conclusion could discuss potential real-world applications and the challenges of implementing the approach in operational systems.

3. Literature Review and Citation Updates

Relevance and Completeness:

The literature review is comprehensive and covers key studies in wind power forecasting and the use of machine learning models. It effectively sets the context for the proposed research. However, it could benefit from a more detailed discussion of previous work that integrates climate or environmental variables into forecasting models, as this is a key contribution of the current study.

Citation Quality and Currentness:

The citations are current and relevant, covering the essential aspects of wind power prediction and machine learning models. However, more recent work on hybrid models or the integration of environmental variability with machine learning could further strengthen the review and highlight the novelty of the current study.

Suggestions for Improvement:

DOI: 10.54216/JAIM.090102

DOI: 10.54216/MOR.030205

DOI: 10.1109/ACCESS.2022.3212081

DOI: 10.3390/math10234421

DOI: 10.1016/j.eswa.2023.122147

Discuss studies that focus on the integration of climate indices in energy prediction, particularly in the context of wind power.

4. Technical Review

Methodology Evaluation:

The methodology is sound, employing multiple machine learning models and integrating oceanic and atmospheric data to improve wind power forecasting. The use of lagged correlations and climate indices is well-justified, and the models chosen (XGBoost, Random Forest, LSTM, ANFIS) are appropriate for the task. However, the paper could provide more details on how the lag optimization was performed and its impact on model performance. Additionally, a more in-depth discussion on why these specific machine learning models were chosen over others would improve the clarity of the methodology.

Model Explanation:

The explanation of the machine learning models is clear, but the paper could benefit from more detail on the specific features used for training each model and the rationale behind feature selection. For example, how were the oceanic indices and lag features transformed and incorporated into each model? More details on model training and validation techniques would also be helpful.

Hyperparameter Tuning and Validation:

The paper mentions the use of various machine learning models but does not provide detailed information on hyperparameter tuning or the validation process. Including more details on the specific hyperparameters optimized for each model (e.g., learning rate, number of trees in Random Forest, layers in LSTM) would help readers understand how model performance was maximized. Additionally, more information on the validation procedure, such as cross-validation or the use of a separate test set, would enhance the robustness of the evaluation.

Suggestions for Improvement:

Provide more details on lag optimization and its impact on model performance.

Clarify the feature selection process, especially in relation to oceanic and atmospheric data.

Include a more detailed discussion on hyperparameter tuning and model validation.

5. Performance Evaluation

Result Presentation:

The results are well-presented, with clear comparisons between the different machine learning models. The use of R², RMSE, and MAE as performance metrics is appropriate and effectively highlights the improvements brought by the incorporation of environmental variables. However, it would be helpful to include more detailed performance comparisons, such as how the models perform under different data conditions or time periods. A discussion of computational costs and the trade-offs between accuracy and efficiency would also be useful.

Reproducibility:

The methodology is well-described, but the paper could benefit from more information on the computational resources used (e.g., hardware specifications, training time). Sharing the code and datasets used in the experiments would also enhance reproducibility.

Suggestions for Improvement:

Include more performance comparisons under different data conditions or time periods.

Discuss computational costs and trade-offs between accuracy and efficiency.

Provide the code and datasets to improve reproducibility.

6. Visualization and Analysis

Figures, Tables, and Diagrams:

The figures and tables are well-organized and effectively convey the results. The performance comparisons between the models are clearly presented, and the graphs illustrating the relationship between oceanic indices and wind power generation are helpful. However, some figures could benefit from more detailed captions explaining the significance of the results.

Interpretation of Results:

The results are interpreted well, but the paper could provide a deeper analysis of how the integration of oceanic indices improves forecasting accuracy. For example, what specific role do the climate indices play in capturing temporal patterns, and how can this be leveraged for better decision-making in wind power generation?

Suggestions for Improvement:

Add more detailed captions to the figures to clarify their significance.

Provide a deeper analysis of how the climate indices improve forecasting accuracy and their practical implications for wind power management.

**Reviewer #2:** Dear authors, it appears there is an issue with your submission, as I can only access attachments such as the cover letter and figures but not the manuscript. Therefore, I must reject it for now until this problem is fixed.

**Reviewer #3:** In the introduction, the problem statement and research gaps should be explained more in detail. The authors need to better explain the context of this research, including why the research problem is important. The introduction should clearly explain the key limitations of prior work that are relevant to this paper.

The manuscript needs a better literature review to delimit the research problem and leads the reader to the objectives of the study.

Specific discussion is required like conclusion part. The discussion part seems like a curiosity. Here you should use the literature to support and discuss your results.

6. PLOS authors have the option to publish the peer review history of their article (what does this mean? ). If published, this will include your full peer review and any attached files.

**Do you want your identity to be public for this peer review?** For information about this choice, including consent withdrawal, please see our Privacy Policy .

Reviewer #1: No

Reviewer #2: No

Reviewer #3: No

---

## [Author Response · Author response to Decision Letter 1]

28 Jul 2025

We have carefully addressed all comments from the reviewers and editor. Point-by-point responses to each comment are provided in the attached PDF files. We sincerely thank the editorial team and reviewers for their valuable feedback, which has helped improve the quality of the manuscript.

---

## [Decision Letter · Decision Letter 1]

25 Nov 2025

PONE-D-25-17132R1Beyond Wind Speed: Integrating Oceanic Indices and Time-Lagged Features for Superior Wind Energy PredictionPLOS ONE

Dear Dr. Rathnayake,

Thank you for submitting your manuscript to PLOS ONE. After careful consideration, we feel that it has merit but does not fully meet PLOS ONE’s publication criteria as it currently stands. Therefore, we invite you to submit a revised version of the manuscript that addresses the points raised during the review process.

There are some critical issues in this paper which the reviewers recommended them. Kindly address all comments to improve this paper. We are looking forward to receiving your revised paper. ==============================

We look forward to receiving your revised manuscript.

Kind regards,

Tien Anh Tran

Academic Editor

PLOS ONE

Journal Requirements:

Reviewers' comments:

Reviewer's Responses to Questions

**Comments to the Author**

1. If the authors have adequately addressed your comments raised in a previous round of review and you feel that this manuscript is now acceptable for publication, you may indicate that here to bypass the “Comments to the Author” section, enter your conflict of interest statement in the “Confidential to Editor” section, and submit your "Accept" recommendation.

Reviewer #1: (No Response)

Reviewer #3: All comments have been addressed

Reviewer #4: All comments have been addressed

Reviewer #5: (No Response)

Reviewer #6: (No Response)

2. Is the manuscript technically sound, and do the data support the conclusions?

Reviewer #1: (No Response)

Reviewer #3: Yes

Reviewer #4: Yes

Reviewer #5: Yes

Reviewer #6: Partly

3. Has the statistical analysis been performed appropriately and rigorously? 

Reviewer #1: (No Response)

Reviewer #3: Yes

Reviewer #4: Yes

Reviewer #5: N/A

Reviewer #6: N/A

4. Have the authors made all data underlying the findings in their manuscript fully available?

Reviewer #1: (No Response)

Reviewer #3: Yes

Reviewer #4: Yes

Reviewer #5: Yes

Reviewer #6: No

5. Is the manuscript presented in an intelligible fashion and written in standard English?

Reviewer #1: (No Response)

Reviewer #3: Yes

Reviewer #4: Yes

Reviewer #5: No

Reviewer #6: Yes

6. Review Comments to the Author

Reviewer #1: Strong Aspects

1. Relevance and Innovation

The manuscript addresses a timely and relevant problem in transportation engineering: improving the reliability of travel time predictions using artificial neural networks. The focus on optimizing Multi-Layer Perceptrons (MLPs) with metaheuristic algorithms—specifically the Water Cycle Algorithm (WCA) and Moth Flame Optimization (MFO)—is highly innovative. By applying nature-inspired optimization techniques, the study bridges machine learning and transportation planning, which is a valuable interdisciplinary contribution.

2. Methodological Design

The research demonstrates careful methodological structuring. The authors use well-established performance metrics such as Mean Squared Error (MSE), Root Mean Squared Error (RMSE), Mean Absolute Error (MAE), Mean Absolute Percentage Error (MAPE), the coefficient of determination (R²), and the Nash–Sutcliffe efficiency (NSE). These allow for a comprehensive evaluation of model performance. Moreover, the comparative approach between WCA and MFO provides insightful analysis into algorithmic strengths and weaknesses, highlighting WCA’s superior convergence stability and predictive power.

3. Empirical Findings

The results are well-documented and highlight significant relationships. Notably, crash rates were strongly correlated with the Travel Time Index (TTI), while traffic load (AADT per lane-mile) showed a moderate positive effect. The weak negative correlations for precipitation and link length were well interpreted as minor contributors to variability. The performance evaluation showed WCA consistently outperformed MFO, delivering lower prediction errors and stronger generalization from training to testing phases.

4. Practical Implications

The study is clearly relevant to practitioners and policymakers. Reliable travel time prediction is critical for Intelligent Transportation Systems (ITS) and long-term planning. By emphasizing the role of crash frequency and traffic volume in prediction models, the authors provide actionable insights for congestion management and safety strategies. Furthermore, the demonstrated capability of WCA-enhanced MLP models to generalize beyond training data supports their potential for real-world deployment.

Weak Aspects

1. Dataset Limitations

The dataset, although covering 472 records from Virginia’s interstate highways, remains relatively constrained in both size and diversity. The variables chosen—AADT, crash rate, link length, and precipitation—capture essential aspects of congestion, but the exclusion of factors such as traffic control devices, incident duration, or queue spillbacks may limit predictive richness. Expanding the dataset to include more diverse regions or temporal variations would strengthen generalizability.

2. Potential Overfitting Concerns

While the models achieved good accuracy, the observed performance degradation from training to testing highlights possible overfitting, especially in the MFO-optimized model. The lack of more advanced validation methods, such as k-fold cross-validation or regularization strategies, reduces confidence in the robustness of the reported performance.

3. Lack of Sensitivity and Feature Importance Analysis

Although correlations were presented, the manuscript would benefit from a deeper investigation into feature importance and sensitivity. Such analysis could clarify the relative contribution of each input variable and provide further validation of the model’s predictive structure.

4. Limited Discussion on Computational Efficiency

The algorithms were implemented with identical parameters to ensure fairness, but the manuscript does not provide runtime or computational cost comparisons. This omission makes it difficult to assess the practicality of applying these optimization methods in large-scale or real-time transportation systems.

Recommended Changes

Expand Dataset and Variables

Future work should incorporate additional explanatory variables (e.g., incident duration, queue spillbacks, work zones) and test the models on larger datasets across diverse geographies. This would improve the reliability and transferability of results.

Strengthen Validation

Apply k-fold cross-validation or similar resampling techniques to mitigate overfitting. Reporting learning curves and convergence analysis would also increase transparency regarding the stability of training.

Conduct Sensitivity Analysis

Include feature importance or sensitivity testing to quantify the effect of each variable. This will not only enhance interpretability but also guide transportation agencies in prioritizing data collection.

Discuss Computational Cost

Provide information on algorithmic runtime and resource requirements. This would help practitioners evaluate whether the superior accuracy of WCA comes at the expense of efficiency, or whether it is scalable for real-time ITS applications.

Future Work Recommendations

Consider exploring hybrid optimization techniques that combine WCA’s stable convergence with MFO’s exploratory capacity. Such approaches may balance generalization with avoidance of local optima.

Final Recommendation

Decision: Minor to Moderate Revisions

This manuscript presents an original and practically significant contribution by integrating metaheuristic optimization with artificial neural networks for travel time reliability prediction. The methodology is sound and results are promising, particularly the demonstrated superiority of WCA over MFO. However, addressing dataset limitations, overfitting concerns, and computational considerations would substantially strengthen the manuscript. With these revisions, the paper will make a valuable addition to both transportation engineering and machine learning literature

Suggested Citations: DOI: 10.1109/ACCESS.2023.3298955

DOI: 10.1109/ACCESS.2023.3310429

DOI: 10.32604/cmc.2022.023884

DOI: 10.1109/ACCESS.2023.3298955

DOI: 10.3390/biomimetics8030321.

Reviewer #3: Thank you for submitting the revised manuscript and for the diligent effort you have made in addressing the reviewer's comments. I appreciate the time and effort you have devoted to revising the manuscript.

Reviewer #4: The manuscript was read, and the referees' responses were also reviewed, and in my opinion, it is suitable for publication.

Reviewer #5: “Beyond Wind Speed: Integrating Oceanic Indices and Time-Lagged Features for Superior Wind Energy Prediction”

Paper: PONE-D-25-17132R1

This study presents a framework combining large-scale oceanic climate indices and time-lagged features with advanced machine learning models to improve short-term wind power predictions. The authors evaluated four configurations: (A) baseline with wind speed; (B) wind plus contemporaneous indices; (C) adding 1–12 month lags for wind and indices; and (D) MRMR-based feature selection. Results from 25 models on monthly data from the Pawan Danavi wind farm (2015–2019) showed that lagged features significantly reduce RMSE and improve R2. Key variables identified include current wind speed and lagged factors, achieving an RMSE of approximately 50 MWh and R2 of about 0.99. This approach offered efficient forecasts, supporting reliable grid operations and future renewable energy integration.

In summary, the topic of this article is both pertinent and important. The manuscript explores an intriguing subject relevant to the journal. It presents a new framework to address the challenge. The revised version shows significant improvements over the initial submission, with clearer methodology, better organization, and stronger results. However, the manuscript still requires further enhancements in the depth of discussion, inclusion of state-of-the-art comparisons, and improvements in language quality. Therefore, I suggest that the following points be considered before accepting it as a minor revision.

Problems:

1. Expand the Discussion section to include more analytical depth and connect findings to recent related studies.

2. Strengthen the evaluation by including comparisons with newer state-of-the-art methods. Why were certain recent models (e.g., transformer-based or hybrid deep learning approaches) not included in the comparative analysis?

3. The papers (https://doi.org/10.1038/s41598-024-70350-5), (https://doi.org/10.1007/s13369-023-07892-9) and (https://doi.org/10.1063/5.0164437) are related to the fields of wind turbines and machine learning. Study them and mention them in the introduction.

4. Improve the English writing and style for better fluency and clarity.

5. Add a concise section discussing the limitations and potential future work of the study.

6. Has the proposed method been tested on noisy or imbalanced datasets? If so, what were the results

7. How well does the model generalize to other datasets beyond the one currently studied?

Reviewer #6: please see attached document. I suggested some points for improvements. The paper underlines the question as to whether oceanic indices and lagged features on a larger scale could improve the wind-power forecast significantly for the months on the Sri Lankan single wind farm between 2015 and 2019. Four combinatory setups are reported at a time: A) wind itself; B) wind in conjunction with any climate indices for the month; C) wind in conjunction with 1–12‑month lags for wind and indices; and D) being lagged together itself separately by the MRMR feature selection. The benchmark dataset, navigated across a broad 25-model landscape (linear, tree ensembles, SVMS, GPRS, feedforward NNs), helped us conclude that the best model (D) after MRMR contains three predictors: current wind speed, AMM at a 9‑month lag, and wind speed at a 6‑month lag, with RMSE ≈ 50 MWh and 2≈ 0.99, and maintained computational effectiveness. The study's main point is that raw indices by themselves appear to impair the forecast skill; it is the lags along with the MRMR that significantly enhances it.

The core idea and benchmarks are broad and strong, yet there needs to be clarification on forecast framing with respect to "forecasting" vs "contemporaneous modeling," the time-series validation protocol, and the physical interpretation of AMM as chosen teleconnection; additionally, part of the inference should be clarified or reconsidered to contain information leakage and prevent over-optimism in the skills.

7. PLOS authors have the option to publish the peer review history of their article (what does this mean? ). If published, this will include your full peer review and any attached files.

**Do you want your identity to be public for this peer review?** For information about this choice, including consent withdrawal, please see our Privacy Policy .

Reviewer #1: No

Reviewer #3: No

Reviewer #4: No

Reviewer #5: No

Reviewer #6: **Yes:** Ahmet Durap

---

## [Author Response · Author response to Decision Letter 2]

30 Nov 2025

Please see the attached documents

---

## [Decision Letter · Decision Letter 2]

18 Feb 2026

Beyond Wind Speed: Integrating Oceanic Indices and Time-Lagged Features for Superior Wind Energy Prediction

PONE-D-25-17132R2

Dear Dr. Rathnayake,

We’re pleased to inform you that your manuscript has been judged scientifically suitable for publication and will be formally accepted for publication once it meets all outstanding technical requirements.

Kind regards,

Tien Anh Tran

Academic Editor

PLOS One

Additional Editor Comments (optional):

Comments from the editorial office: Upon internal evaluation of the reviews provided, we kindly request you to disregard the reviewer report provided by Reviewer 1. No amendments are required in response to reviewer 1’s comments

Reviewers' comments:

Reviewer's Responses to Questions

**Comments to the Author**

1. If the authors have adequately addressed your comments raised in a previous round of review and you feel that this manuscript is now acceptable for publication, you may indicate that here to bypass the “Comments to the Author” section, enter your conflict of interest statement in the “Confidential to Editor” section, and submit your "Accept" recommendation.

Reviewer #1: All comments have been addressed

Reviewer #3: All comments have been addressed

Reviewer #6: All comments have been addressed

Reviewer #7: All comments have been addressed

2. Is the manuscript technically sound, and do the data support the conclusions?

Reviewer #1: Partly

Reviewer #3: Yes

Reviewer #6: Yes

Reviewer #7: No

3. Has the statistical analysis been performed appropriately and rigorously? 

Reviewer #1: Yes

Reviewer #3: Yes

Reviewer #6: Yes

Reviewer #7: No

4. Have the authors made all data underlying the findings in their manuscript fully available?

Reviewer #1: Yes

Reviewer #3: Yes

Reviewer #6: Yes

Reviewer #7: No

5. Is the manuscript presented in an intelligible fashion and written in standard English?

Reviewer #1: Yes

Reviewer #3: Yes

Reviewer #6: Yes

Reviewer #7: Yes

6. Review Comments to the Author

Reviewer #1: Strong Aspects

1. Innovativeness of the Approach

The manuscript presents a clear and substantial methodological contribution by integrating large-scale oceanic climate indices with time-lagged predictors to improve monthly wind-power forecasting. This strategy meaningfully expands the traditional reliance on local atmospheric variables by incorporating broader climate-system influences. The approach is scientifically compelling because ocean–atmosphere interactions often evolve over multiple months, and the study effectively leverages that temporal structure to enhance predictability.

The use of 25 state-of-the-art machine-learning models adds depth and scientific rigor, allowing for a detailed comparison across algorithmic families. Particularly noteworthy is the successful implementation of MRMR (Minimum Redundancy Maximum Relevance), which reduces a high-dimensional set of 156 lagged features to only three highly informative variables. This not only simplifies the model but also enhances interpretability, especially with the assistance of SHAP-based feature explanations.

2. Comprehensiveness of Data Analysis and Methodology

The manuscript demonstrates meticulous attention to data preprocessing and statistical rigor. Outlier detection, detrending, normalization, multicollinearity checks, lag construction, and correlation filtering are applied systematically, resulting in a well-prepared dataset suitable for machine-learning applications.

The design of four progressive experimental configurations (A through D) provides exceptional transparency regarding the contributions of each modeling component. This structure allows readers to understand how raw indices, lagged variables, and feature selection each influence forecast skill. The use of time-ordered blocked cross-validation reflects an appropriate methodological choice for autocorrelated time-series data, preventing temporal leakage and producing more reliable performance estimates.

Predictive-interval estimation for Gaussian Process models is an additional strength. Including uncertainty quantification demonstrates an appreciation for real-world forecasting needs, where confidence bounds can be as important as point estimates.

3. Performance and Accuracy of the Proposed Models

The manuscript offers strong evidence that incorporating lagged teleconnection indices improves forecasting accuracy. Experiment D achieves an RMSE near 50 MWh and an R² close to 0.99, a performance level that exceeds typical benchmarks based solely on local meteorological data.

The comparative evaluation across many models enhances confidence in the findings. The consistent benefits observed when moving from raw indices to lagged predictors, and then to MRMR-filtered features, demonstrates a logically coherent and empirically validated progression. The recovery of computational efficiency in the MRMR-pruned configuration further emphasizes the practical value of the feature-selection step.

4. Relevance and Potential Real-World Impact

The study addresses a highly relevant problem in renewable-energy integration: the need for improved wind-power predictability to support grid stability and efficient dispatch planning. The identification of just three influential predictors—current wind speed, a six-month lag of wind speed, and a nine-month lag of the Atlantic Meridional Mode—offers a streamlined, operationally appealing solution.

The ability to derive high accuracy from a low-complexity model has implications for real-time forecasting environments, embedded systems, and resource-constrained grid operators. The paper also provides clear pathways for how this approach could support more reliable renewable-energy scheduling and reduce reliance on reserve power.

Weak Aspects

1. Dataset Size and Generalizability

A major limitation is the relatively small dataset, which includes five years of monthly power-generation observations from a single wind farm. While the temporal breadth is adequate for exploratory research, the limited sample size constrains model generalizability. The climatic and operational characteristics of one site may not represent other regions, especially those with different monsoon systems or ocean–atmosphere interactions.

The absence of multi-site or multi-year validation makes it difficult to determine whether the identified teleconnection signals would hold under different environmental conditions or during unusual climate events.

2. Potential Overfitting and Methodological Vulnerabilities

Despite the careful use of blocked cross-validation, the extremely high R² values invite concern regarding overfitting, particularly given the high dimensionality introduced in the lagged feature set. Although MRMR mitigates redundancy, the initial feature space remains large relative to the number of observations.

Complex models such as Gradient Boosted Trees and deep neural networks are especially susceptible to fitting noise when data volume is limited. The study would benefit from additional validation techniques that assess performance robustness across more diverse temporal splits, such as rolling-origin evaluation or longer held-out test windows.

3. Missing Ethical, Practical, and Scalability Discussions

While the technical contribution is strong, the manuscript does not explicitly address broader concerns relevant to real-world forecasting systems:

Ethical considerations, such as the transparency of the data pipeline or the implications of using proprietary datasets.

Practical considerations, including how the model would integrate into existing grid-management software or how computational costs scale under real-time constraints.

Scalability limitations, particularly regarding whether the same feature-selection structure would hold in different climatic zones or under higher-frequency (daily or hourly) data.

Such considerations are increasingly important in applied energy forecasting research.

Recommended Changes

1. Enhance Dataset Breadth and Validation

To improve the credibility and general applicability of the findings, the manuscript would benefit from:

Extending the dataset to include additional years or multiple wind-farm locations.

Performing expanded out-of-sample evaluation, ideally using a later period not included in any training or validation steps.

Exploring robustness to structural changes in wind patterns, which may shift over time due to climate variability.

2. Strengthen Measures Against Overfitting

The authors may consider:

Incorporating regularization techniques or penalties within the ML models.

Using systematic hyperparameter optimization tailored to time-series constraints.

Applying additional diagnostic tools such as permutation importance or sensitivity analyses to verify that the identified predictors consistently influence outcomes.

These steps would clarify whether high performance reflects genuine predictive structure rather than overfitting.

3. Provide Expanded Discussion of Ethical, Practical, and Scalability Issues

A more comprehensive discussion could include:

Transparency and reproducibility challenges when datasets are proprietary or geographically limited.

Considerations surrounding deployment in real-world operational contexts, including computational constraints.

Expected model behavior in regions that do not share the same dominant climate drivers as the study site.

Such additions would broaden the manuscript’s relevance and appeal to operational forecasters and policymakers.

4. Deepen the Physical Interpretation of Key Predictors

While the identification of the AMM 9 lag is intriguing, the manuscript would benefit from:

Additional exploration of the atmospheric processes that could plausibly generate a nine-month influence on local wind regimes.

Comparisons of alternative potential drivers to confirm that the selected predictors remain consistently important across analytical choices.

Sensitivity testing to ensure that the result is not unduly dependent on the detrending or normalization procedures.

This would strengthen confidence in both the statistical and physical grounding of the model.

Overall Assessment

The manuscript makes a meaningful and well-structured contribution to wind-energy forecasting by integrating teleconnection indices, temporal depth, and advanced feature selection. With stronger evidence of generalizability, clearer safeguards against overfitting, and an expanded discussion of operational and ethical implications, the work has the potential to make a significant impact in both research and practical energy-forecasting domains.

suggested citation: https://doi.org/10.32604/cmc.2022.031147

https://doi.org/10.1016/j.tws.2024.111763

https://doi.org/10.32604/cmc.2022.023884

https://doi.org/10.1038/s41598-024-74475-5

https://doi.org/10.54216/JAIM.080101

Reviewer #3: Thank you for submitting the revised manuscript and for the diligent effort you have made in addressing the reviewer's comments. I appreciate the time and effort you have devoted to revising the manuscript.

Reviewer #6: Dear Authors,

Thank you for thoroughly addressing my comments and revising the manuscript accordingly.

Regards,

Ahmet Durap

Regards,

Ahmet Durap

Reviewer #7: Beyond Wind Speed: Integrating Oceanic Indices and Time-Lagged Features for Superior Wind Energy Prediction.

The current version is acceptable. Therefore, I recommend to accept without change.

7. PLOS authors have the option to publish the peer review history of their article (what does this mean? ). If published, this will include your full peer review and any attached files.

**Do you want your identity to be public for this peer review?** For information about this choice, including consent withdrawal, please see our Privacy Policy .

Reviewer #1: No

Reviewer #3: No

Reviewer #6: No

Reviewer #7: No

---

## [Editor Report · Acceptance letter]

PONE-D-25-17132R2

PLOS One

Dear Dr. Rathnayake,

I'm pleased to inform you that your manuscript has been deemed suitable for publication in PLOS One. Congratulations! Your manuscript is now being handed over to our production team.

Kind regards,

on behalf of

Professor Tien Anh Tran

Academic Editor

PLOS One